f

# Enhancing the operational value of snowpack models with visualization design principles

Simon Horton[1], Stan Nowak[1], and Pascal Haegeli[1]

[1]Simon Fraser University, Burnaby, BC, Canada

**Correspondence:** Simon Horton (shorton@avalanche.ca)

**Abstract.** Forecasting snow avalanches requires a reliable stream of field observations, which are often difficult and expensive to collect. Despite the increasing capability of simulating snowpack conditions with physical models, models have seen limited adoption by avalanche forecasters. Feedback from forecasters suggest model data is presented in ways that are difficult to interpret and irrelevant to operational needs. We apply a visualization design framework to enhance the value of snowpack models to avalanche forecasters. An established risk-based avalanche forecasting workflow is used to define the ways forecasters solve problems with snowpack data. We suggest model data should be visualized in ways that directly support common forecasting tasks such as identifying snowpack features related to avalanche problems and locating avalanche problems in terrain at relevant spatial scales. Examples of visualizations that support these tasks and follow established perceptual and cognitive principles from the field of information visualization are presented. Interactive designs play a critical role in understanding these complex datasets and are well suited for forecasting workflows. Although extensive user testing is still needed to evaluate the effectiveness of these designs, visualization design principles open the door to more relevant and interpretable applications of snowpack model for avalanche forecasters. This work sets the stage for implementing snowpack models into visualization tools where forecasters can test their operational value and learn their capabilities and deficiencies.

## 1 Introduction

Numerical environmental and weather prediction models have dramatically transformed the accuracy of weather forecasts and the role of weather forecasters since the 1980s (Benjamin et al., 2019). As model performance improved, forecasting tasks shifted from predicting weather conditions to interpreting and communicating model guidance. A centerpiece in the adoption of prediction models by weather forecasters was the development of visualization tools that allowed them to work directly with gridded modelled data in combination with in-situ weather observations and remote sensing data (Benjamin et al., 2019). This setup allowed forecasters to visualize model output along with observations and gradually learn the operational value of the models.

The work of avalanche forecasters is similar in nature and complexity to the work of weather forecasters. The objective of avalanche forecasting is to develop an accurate mental model of the current and future nature of avalanche hazard by integrating avalanche, snowpack, and weather information from a variety of sources (Canadian Avalanche Association, 2016b).

This assessment is then combined with terrain information to make risk management decisions regarding specific elements at risk. The spatial scale of avalanche forecasting can range from individual slopes in backcountry guiding, to groups of avalanche paths when protecting infrastructure, to entire mountain ranges in public avalanche warnings. To assist avalanche forecasters at the higher end of the spatial scale spectrum, physical snowpack models such as Crocus (Brun et al., 1992) and SNOWPACK (Lehning et al., 1999) were developed in the 1990s to provide supplementary data about snowpack conditions. Despite the fact model developers have created numerous operational tools to visualize model output, snowpack models have so far only seen limited adoption into operational workflows when compared to weather prediction models (Morin et al., 2020).

Morin et al. (2020) employed the information quality framework of Bovee et al. (2003) to describe issues with operational snowpack model tools in terms of the accessibility, interpretability, relevance, and integrity of the information. Accessibility to snowpack models is limited by the time constraints in forecasting environments and workflows that are designed for field data rather than model data. Existing tools are also difficult to interpret as model output is complex and in their current form require expertise or substantial training to comprehend and apply. The relevance of the information they provide is also questioned, as similar information may be available from other sources. The integrity of model output is also difficult to evaluate in an operational setting where there is limited validation data. Snowpack models can produce snow stratigraphy profiles for multiple parameters (e.g. grain size, hardness, temperature) at different time intervals at potentially hundreds or thousands of locations. This data can be so complex and voluminous that it becomes challenging for operational forecasters to make sense of in its raw form using conventional methods such as viewing manual snow stratigraphy profiles. This has been described as "information overload" and characterizes "big data" environments (De Mauro et al., 2016). As avalanche forecasting requires substantial cognitive effort to continuously maintain a mental model of conditions (Maguire and Percival, 2018), introducing additional complex data can disrupt this process and have adverse effects on performance. Based on their analysis, Morin et al. (2020) aptly conclude that while it was important for researchers to focus on improving the accuracy of snowpack models, we are now at a point where addressing issues with the design of operational tools is critical for making snowpack models truly valuable for avalanche forecasting.

To address the challenges of big data and make it tractable for human analysis, the field of visual analytics blends automatic analysis with human analysis via visual interfaces (Keim et al., 2008). Specifically, visualization in combination with interaction techniques support a process of iterative inquiry into data to support sense-making. This reduces the cognitive work needed to perform analytic tasks by leveraging the pattern detecting abilities of the human visual system for processing complex information that would normally exceed cognitive limits (Ware, 2012). Visual analytics has made complex problems and model output tractable for non-scientists and non-model experts in a variety of domains including physics, business, intelligence analysis, and disaster management (Keim et al., 2008). Effective visualization techniques are particularly valuable for environmental data, which is often complex due its spatiotemporal dimensions and uncertainties (Grainger et al., 2016). For example, studying visualization design principles has improved the interpretability of complex data sets in the fields of meteorology (Rautenhaus et al., 2018; Stauffer et al., 2015) and oceanography (Thyng et al., 2016).

Judging from the success of visual analytics applications in other disciplines of environmental science, we believe applying a visualization design perspective to snowpack models has the potential to substantially address some of the shortcomings that

have so far limited their operational use. In this paper, we present design principles for visualization tools that increase the interpretability and relevance of snowpack models for operational avalanche forecasters. These design principles are informed by information visualization, avalanche forecasting practices, and the unique features of snowpack model data. First, we apply a visualization design framework to the domain of avalanche forecasting to outline principles of how data should be visualized to solve operational problems (Sect. 2). Then we provide examples of visualizations where these principles are applied with snowpack model data (Sect. 3), followed by suggestions for next steps towards operational applications (Sect. 4) and conclusions (Sect. 5).

## 2 Visualization design principles for avalanche forecasting

### 2.1 Nested levels of visualization design

The nested model for visualization design described by Munzner (2009) has established itself as a valuable framework for designing and evaluating visualization tools. This framework considers four nested levels where distinct design issues arise, and where issues at one level can cascade to other levels. The issues with operational snowpack model tools identified by Morin et al. (2020) relate to design issues at each level of the nested model. These four levels provide designers with a tangible framework for understanding the users' problems, showing the appropriate information, and presenting it both effectively and efficiently:

1. *Domain situation level.* The domain situation describes the target users, their field of interest, their questions, and their data. A domain has unique vocabulary for describing its data and problems, and usually has an existing workflow for how data is used to solve problems. Issues arise when designers misunderstand the users' needs. For example, existing tools that present snowpack model data may not address the major needs and questions of avalanche forecasters, such as assessing the spatial distribution of an avalanche problem (relevance).

2. *Task and data abstraction level.* Task and data abstraction maps domain-specific problems into generic vocabulary that clearly describe what type of data is being visualized and why. Tasks are described with generic verbs (e.g. locate, compare) and data is described with generic nouns and adjectives (e.g. table, network, ordered, categorical). Issues arise when the functions and data types in a design do not solve the intended problem. For example, detailed snow stratigraphy profiles provided by snowpack models may not be the type of information needed for specific forecasting tasks (relevance).

3. *Visual encoding and interaction idiom level.* This level creates visual representations of the data. A distinct visual representation is called an idiom. Data is encoded by arranging it along spatial dimensions and mapping attributes to non-spatial visual features such as colour, size, and shape, while interaction idioms allow the user to change the view. Issues arise when idioms are ineffective at visualizing information. Existing idioms for visualizing snowpack model data are often complex, busy, and difficult for non-model experts to understand (interpretability).

4. *Algorithm level.* This is the level where idioms are produced from raw data with a computer. Issues arise when algorithms are too slow. At the algorithm level, most snowpack model visualizations are too time consuming for forecasters because they are poorly integrated into their workflows (accessibility).

Munzner (2014) also describes that visualization problems can be attacked from two possible directions within the nested model: top-down approaches that first understand the domain and tasks and then design visual idioms accordingly, and bottom-up approaches that start with developing new algorithms and idioms. Most existing snowpack model visualizations were developed with bottom-up approaches that began with model development followed by the creation of visualizations of the model output. Bottom-up approaches allow novel visualizations that reveal nuances and anomalies in new types of data, but also have the potential to not solve the intended problem (Munzner, 2014). While it is worth considering bottom-up designs that take advantage of the unique capabilities of snowpack models, it is also important to carefully examine the domain and tasks of avalanche forecasting to establish top-down design principles that support forecasting needs.

## 2.2   Domain of avalanche forecasting

Avalanche forecasting is a common task for all operations that manage short-term avalanche risk (e.g. ski areas, transportation corridors, backcountry warnings, resource extraction). The forecasting process consists of iterative data analysis and is dominated by human judgement and inductive logic (LaChapelle, 1980; McClung, 2002). Statham et al. (2018) surveyed existing operational practices within North American avalanche forecasting operations to develop a standard framework for this process. The resulting conceptual model of avalanche hazard (CMAH) identifies the key components of avalanche hazard and provides standard workflow and terminology to guide the forecasting process. The CMAH is a risk-based framework that is consistent with other natural hazard disciplines and can be applied to any scale in space or time. A central part of the CMAH is the concept of avalanche problems that represent individual, identifiable operational concerns that can be described in terms of their potential avalanche type, location, likelihood, and size (Statham et al., 2018). Under the CMAH, avalanche forecasting is viewed as sequentially answering four questions:

1. What type of avalanche problems exist?

2. Where are these problems located in the terrain?

3. How likely is it that an avalanche will occur?

4. How destructive will the avalanche be?

Over the past decade, the CMAH has been widely adopted by all industry sectors in North America (Statham et al., 2018), which clearly indicates that it is a useful model to describe the domain situation of avalanche forecasting.

## 2.3   Task and data abstractions for snowpack analysis

Given the importance of avalanche problems in avalanche forecasting, operational visualization tools should be designed to help forecasters identify and characterize avalanche problems. Assessing avalanche problems consists of integrating a complex

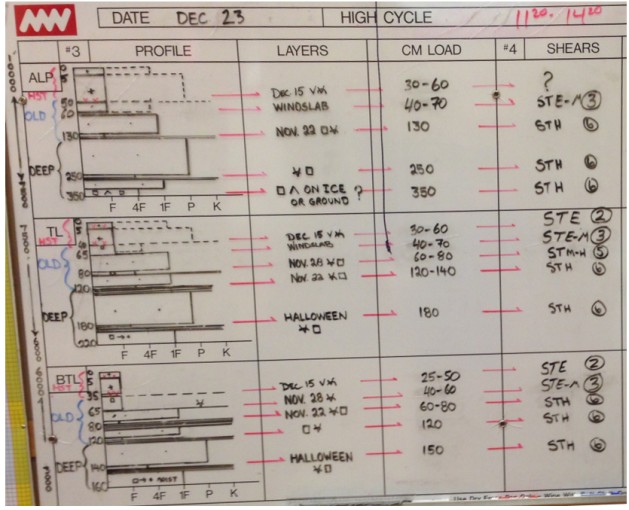

**Figure 1.** Example of a snowpack summary for a large forecast area drawn on a whiteboard to summarize relevant snowpack conditions. This example was drawn on 23 December 2017 by an avalanche forecaster at Mike Wiegele Helicopter Skiing in Canada. Each row shows conditions for a different elevation band (alpine, treeline, and below treeline). The profile column has stratigraphy profiles showing typical layer depths and hardness (widespread layers are shown with solid lines and localized layers with dashed lines). The remaining columns identify important layers and provide details on their burial date, grain type, depth (in cm), and representative shovel shear test results (Photo: Mike Wiegele Helicopter Skiing).

array of data that includes observations of avalanches, snowpack, weather, and terrain (Statham et al., 2018). There is no structured or standardized way this data is used to answer the CMAH questions, as the analysis relies on subjective judgement and heuristics (LaChapelle, 1980), however there are common practices for interpreting field observations.

Snowpack models produce data that is analogous to manual snow stratigraphy profiles, which is a key type of field data used by forecasters to understand snowpack conditions. Building off familiar visual representations is an effective way for people to understand new types of information (Blackwell, 2001), and thus examining existing practices for visualizing and analyzing manual snow profiles provides insight into ways snowpack models could be visualized to support forecasting tasks.

Forecasters perform several analysis tasks with manual snow profiles to develop a comprehensive mental model of hazard conditions. Manual snow profiles are either recorded in tables of unstructured text or illustrated as hardness stratigraphy profiles (Canadian Avalanche Association, 2016a). Forecasters learn to identify relevant snowpack features in these profiles, and then compare multiple snow profiles along with other observations to summarize the snowpack conditions within a forecast area. Forecasters summarize snowpack data in a written snowpack summary that gives a concise overview of conditions in their forecast area. The goal of a written snowpack summary is to organize and reduce data, focusing on average conditions along with potential anomalies and outliers (Canadian Avalanche Association, 2016a). Some forecasters visualize their snowpack summary with a representative profile for their forecast area, which helps them organize and communicate relevant information

(Fig. 1). These visual snowpack summaries are an example of where forecasters already use visualization to help summarize and understand complex information.

Tracking trends in snowpack conditions over time is another common forecasting task, which is most often done with tables of text. Temporal trends in the likelihood and size of avalanches are particularly relevant. For example, the InfoEx forecasting workflow allows forecasters to track weak layers in their forecast area with qualitative summaries of their status and depth each day of the season (Haegeli et al., 2014). Basic snowpack observations are plotted as time series (e.g. daily snowfall at fixed observation sites), but complex data like snowpack structure is rarely visualized temporally.

To help forecasters answer the four key questions about avalanche problems posed by the CMAH, visualizations of snowpack data should help forecasters identify, compare, and summarize snowpack features in their forecast area and highlight trends over time. These specific tasks should be considered when designing tools to visualize either field data or snowpack model data.

## 2.4 Information visualization principles

The field of information visualization studies how to leverage the human visual system to off-load cognitive work and visualize information effectively. Information visualization principles should be considered when designing the visual appearance and interactive components of tools for snowpack model data (i.e. the visual encoding and interaction idiom level of the nested model). These principles consider effective ways of representing data visually and are explained in greater detail in textbooks by Ware (2012) and Munzner (2014). The following list summarizes information visualization principles that are relevant when visualizing snowpack model data:

– When representing information visually, designers encode data to visual features such as: spatial position, size, color, or shape among others. Color can be further divided into hue (the actual color), luminance (the brightness or darkness of a color), and saturation (the intensity of the color). Through years of perception studies, standard guidelines for mapping these visual features to data types have been established (Cleveland and McGill, 1984).

– Visual encodings should present data in ways that match the capabilities of our visual system. Hence, categorical and ordered data should be encoded with visual features that match human visual aptitudes. For example, when using colours, hues should be used for categorical attributes such as avalanche problem types and luminance (lightness or brightness) should be used for ordered attributes such as avalanche likelihood.

– Designs should prioritize the importance of information and encode data to visual features that are perceived more quickly, accurately, and draw our attention to make this information more salient (i.e. noticeable) and discriminable (Cleveland and McGill, 1984). Spatial position is perceived the fastest and most accurately, and thus the most important attributes should be encoded by their position in a visualization. After spatial position, designs should consider the hierarchy of salience for non-spatial visual features. For example, size features such as length and area are more salient than colour features such as hue, luminance, and saturation. For a comprehensive breakdown of this hierarchy see Munzner (2014).

- Choose designs that are accessible and effective for common types of colour blindness. For example, red-green colour blindness (deuteranopes) affects roughly $8\%$ of males of European descent (Birch, 2012).

- Interaction reduces cognitive load and helps users understand data by asking questions and performing queries. A practical guideline for designing interaction idioms is the visual information seeking mantra of Shneiderman (1996): "overview first, zoom and filter, then details on demand". The initial visualization should provide an overview of the entire dataset, and then the interactions should allow the users to change the view to see subsets of the data, and then visualize details about features of interest. This design approach offers users a flexible way to explore data, while being able to maintain a sense of context and orientation.

- Comparison tasks are most effective on aligned scales. Furthermore, comparisons of large amounts of data are often more effective when seeing multiple frames in a single side-by-side view rather than changing views over time. The human perceptual system is effective at reading spatial information in parallel, whereas changing views with animations or multiple tabs relies on human memory and results in substantial cognitive load (Ware, 2012).

- Visualization idioms should present data with the smallest number of spatial dimensions, avoiding three-dimensional visualizations and using one-dimensional lists where possible. Displaying three-dimensional data on planar surfaces has numerous issues with depth-perception and over plotting (Ware, 2012).

## 3  Applications of visualization design principles

This section presents examples of how visualization design principles can be applied to enhance the relevance and interpretability of snowpack model data. Rather than presenting the optimal avalanche forecasting tool, these examples show how to apply a top-down approach to design. Each visualization addresses a specific question posed by the CMAH, which can be combined into a single interactive tool that allows sequential question asking. This section starts by introducing each individual visualization then finishes with an example of an interactive forecasting tool that combines them.

The following examples are for Glacier National Park, Canada, a forecast region covering $1354\,\mathrm{km}^2$ of mountainous terrain (Fig. 2). The examples focus on the needs of regional scale avalanche forecasters (considerations for other forecasting contexts are discussed in Sect. 4). The examples use simulated snowpack data for 8 January 2018, as this day had interesting snowpack conditions with considerable avalanche danger at all elevation bands and two common avalanche problems (Parks Canada, 2018): a storm slab problem at all elevations (size 1 to 2 avalanches were possible to likely) and a persistent slab problem at treeline and below treeline elevations (size 1 to 3 avalanche were possible to likely). Appendix A provides additional examples of the visualizations for several other days throughout the 2017-18 season.

Simulated snow profiles were produced by forcing the physical snowpack model SNOWPACK (Lehning et al., 1999) with gridded meteorological data from the Canadian HRDPS numerical weather prediction model (Milbrandt et al., 2016). Numerous configurations of weather inputs and geometries are possible with snowpack models (Morin et al., 2020). The ideal configuration for avalanche forecasting should produce a representative sample of snow profiles that capture the spatial

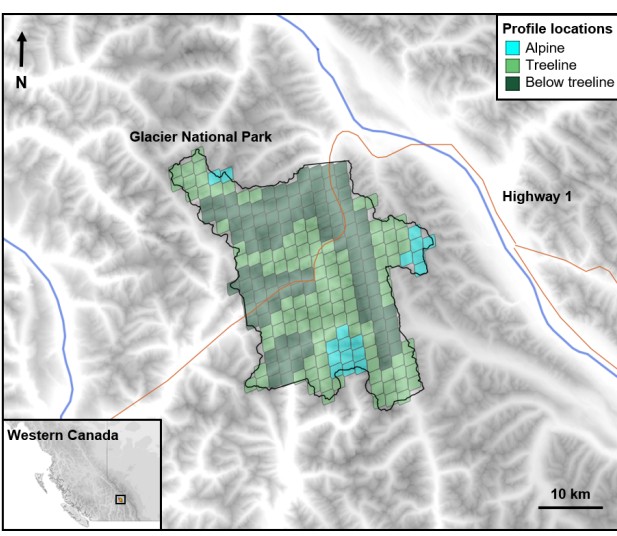

**Figure 2.** Location of Glacier National Park in western Canada, and locations of grid points where meteorological data was extracted to produce simulated snow profiles (coloured according to elevation band).

variability across the forecast region. Choosing an optimal configuration remains an open research question that requires model expertise and field validation. To produce a sample of profiles that cover the type of locations considered by regional forecasters, a gridded approach was used to extract meteorological data from all 236 grid points in the forecast region. A single flat field profile and four virtual slope profiles were simulated at each grid point ($38\,^{\circ}$ slopes in four cardinal directions) resulting in a total of 1180 profiles covering a range of aspect and elevation bands.

## 3.1 Identify snowpack structure patterns with colour

Snowpack features related to avalanche problems should be easy to identify in visualizations of snowpack structure. For example, thin weak layers are important for slab avalanche problems, and so these layers should have high contrast from surrounding layers. From a perceptual perspective, the standard colour palette for snow grains (i.e. Fierz et al., 2009) may cause undesired emphasis on certain types of snow due to the relative contrast between colours. For example, the fuchsia colour used for surface hoar has little contrast with surrounding layers while melt forms and ice formations are highly emphasized (despite being less important for identifying most avalanche problems). The colours also make it difficult for individuals with colour blindness to distinguish important features (e.g. precipitation particles and melt forms are difficult to discern for individuals with red-green colour blindness).

We propose a perception-informed colour palette for snow grain types that emphasizes features related to avalanche problems (Table 1). Similar perception-informed colour palettes have been proposed to improve the interpretation of visualizations in meteorology and oceanography (Stauffer et al., 2015; Thyng et al., 2016). The proposed colour palette groups grain types into four categories based on their role in avalanche problems: persistent weak layers (surface hoar and depth hoar), new snow

**Table 1.** A perception-informed colour palette for snow grain types that emphasizes features related to avalanche problems and is effective in grayscale and for common types of colour blindness (HTML codes provided for perception-informed colour palette).

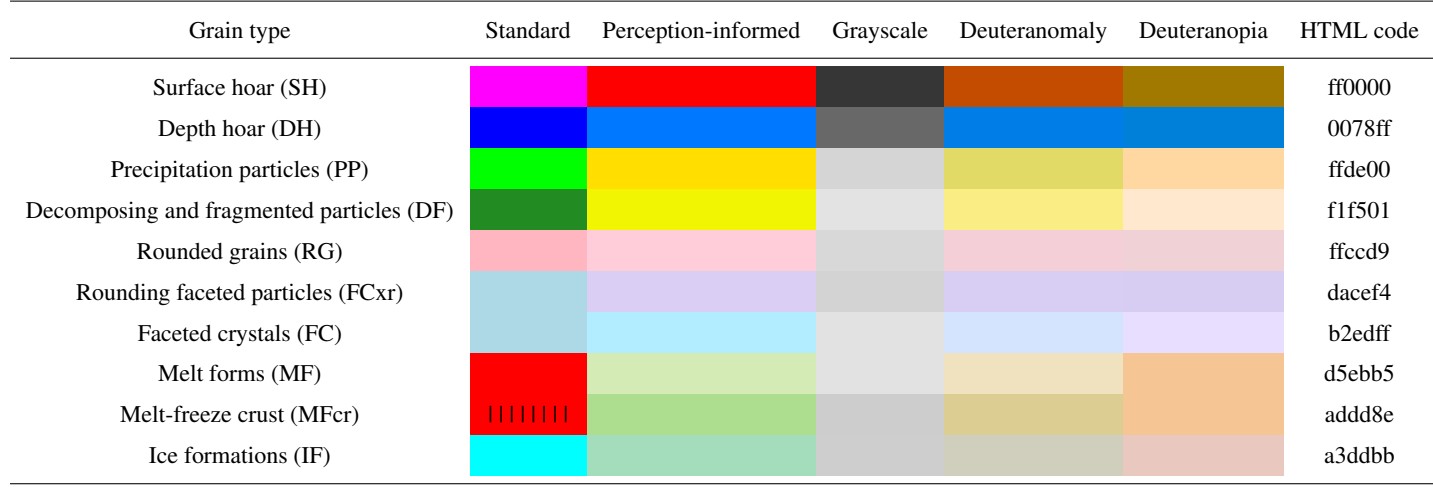

| Grain type | Standard | Perception-informed | Grayscale | Deuteranomaly | Deuteranopia | HTML code |
|---|---|---|---|---|---|---|
| Surface hoar (SH) | | | | | | ff0000 |
| Depth hoar (DH) | | | | | | 0078ff |
| Precipitation particles (PP) | | | | | | ffde00 |
| Decomposing and fragmented particles (DF) | | | | | | f1f501 |
| Rounded grains (RG) | | | | | | ffccd9 |
| Rounding faceted particles (FCxr) | | | | | | dacef4 |
| Faceted crystals (FC) | | | | | | b2edff |
| Melt forms (MF) | | | | | | d5ebb5 |
| Melt-freeze crust (MFcr) | | | | | | addd8e |
| Ice formations (IF) | | | | | | a3ddbb |

**Table 2.** Simplified colour palette for groups of grain types related to avalanche problems.

| Group | Grain types | Colour | HTML code |
|---|---|---|---|
| Persistent weak layers | SH, DH | | 95258f |
| New snow layers | PP, DF | | ffde00 |
| Bulk layers | RG, FCxr, FC | | dacef4 |
| Melt and ice layers | MF, MFcr, IF | | d5ebb5 |

layers (precipitation particles and decomposing and fragmented particles), bulk layers (rounded grains and faceted crystals), and melt and ice form layers. While faceted crystals are typically considered persistent weak layers, the SNOWPACK model classifies any faceted crystal with a grain sizes greater than 1.5 mm as depth hoar. This rule causes most modelled layers composed of large faceted crystals (i.e. those associated with persistent weak layers) to be classified as depth hoar, while layers with smaller faceted crystals tend to be thicker and associated with slabs. While these grain type groups are defined by model behaviour, they are consistent with common snow profile analysis techniques that consider a combination of grain type and grain size (amongst other properties) to identify weak layers (Schweizer and Jamieson, 2007).

These groups were visually related using analogous color schemes (e.g. the hues are perceptually close to each other) that remained visually discriminable. The visual salience of these groups was adjusted using properties of color such as how dark they appear (i.e. luminance) and how vivid the colors are (i.e. saturation). In this way a visual hierarchy of importance was created. Weak layers that tend to take up the smallest area were made the most salient by using strong contrast against other grain types, next new snow was made salient. Finally, the other layers formed the lowest level of perceptual salience and serve as a neutral background. All colors were made to be perceptually distinct and accessible for common types of colour blindness

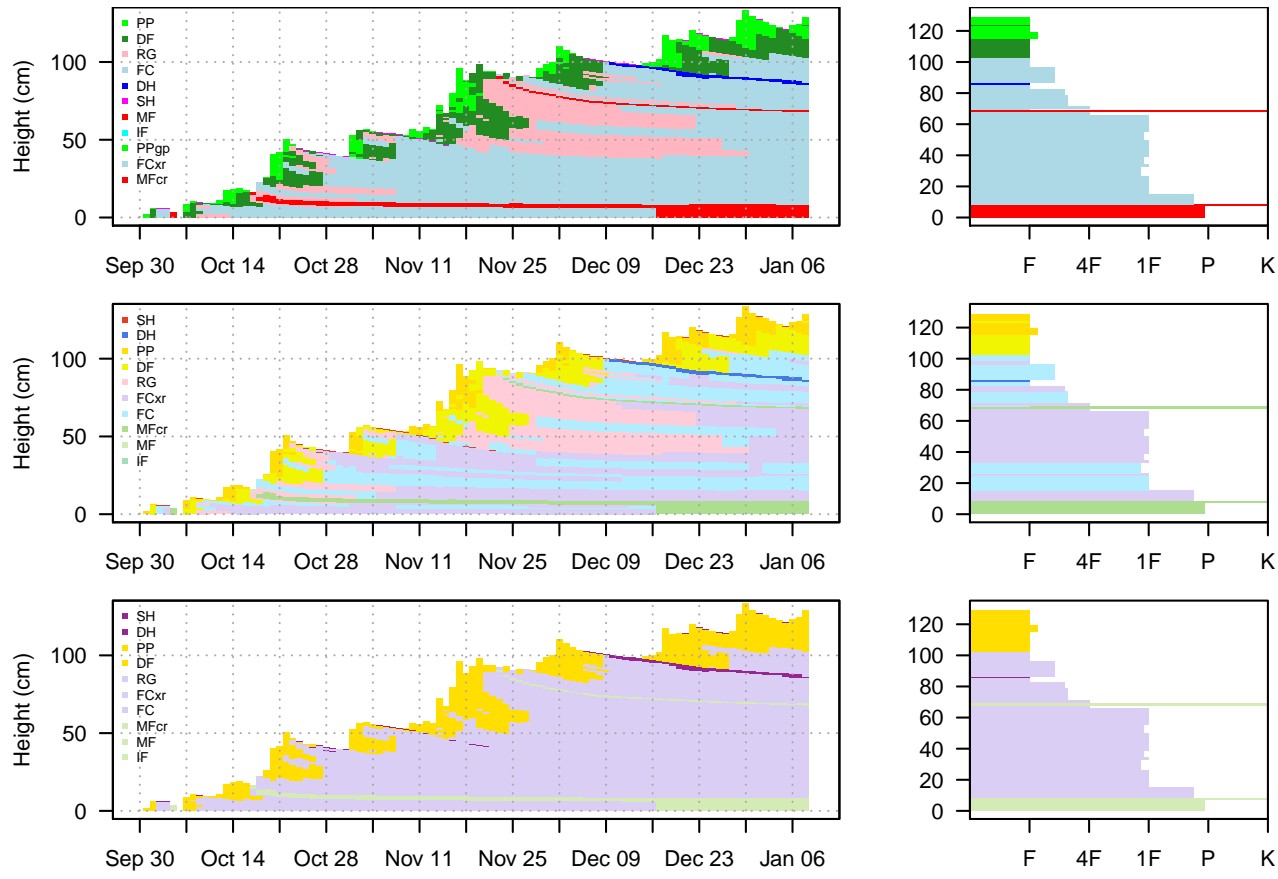

**Figure 3.** Applying different colour palettes to common snowpack model visualizations. The same timeline and stratigraphy profiles are shown with the standard colours for grain types (top row), perception-informed colours for grain types from Table 1 (middle row), and perception-informed colours for grain type groups from Table 2 (bottom row). Timeline profiles (left) show the evolution of layer heights and grain type from 1 October 2017 to 8 January 2018. Stratigraphy profiles (right) show layer height, grain type, and hand hardness (F = fist, 4F = four finger, 1F = one finger, P = pencil, K = knife) on 8 January 2018.

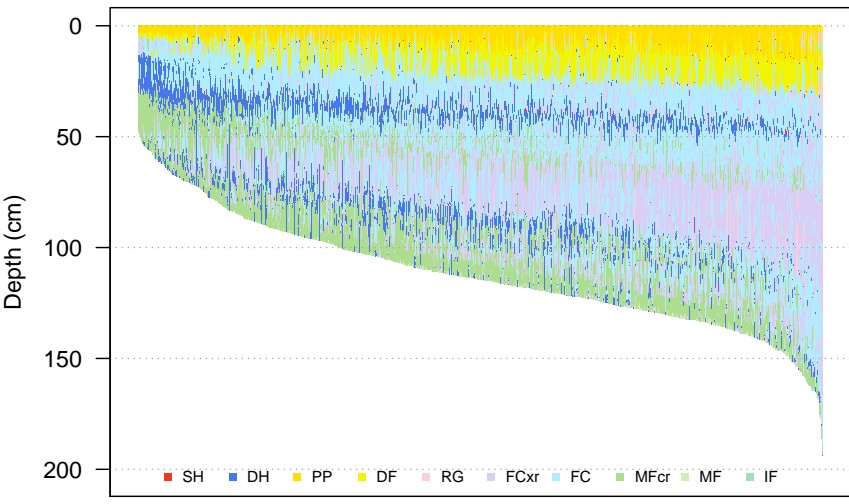

Thinnest snowpack ⟷ Thickest snowpack

**Figure 4.** A visualization to identify avalanche problem types from 1180 simulated profiles on 8 January 2018. The profiles are summarized by plotting grain type stratigraphies side-by-side and sorting the profiles from thinnest to thickest. Grain types are coloured using the perception-informed palette from Table 1. The storm slab avalanche problem is emphasized with yellow surface layers and the persistent slab avalanche problem is emphasized by the band of blue depth hoar layers 30 to 50 cm below the surface.

(see Table 1). Unique colours were also assigned to melt-freeze crust and rounding faceted particles, as distinguishing these sub-classes was deemed meaningful for avalanche forecasters. A simplified colour palette was also designed using only the four main categories of grain types for non-model experts (Table 2). The simplified palette uses colors that are analogous to the full palette and maintain the established visual hierarchy.

The colour palettes were tested with common visualization idioms such as hardness and timeseries profiles (Fig. 3). Comparing the standard and redesigned colour palettes at a single treeline location in Glacier National Park shows how the new palettes simplify the interpretation of the profiles by drawing attention to the most important snowpack features. The increased salience of the thin depth hoar layer highlights a potential persistent slab avalanche problem and the new snow highlights a potential storm slab avalanche problem.

### 3.2 Identify avalanche problem types from multiple profiles

Visualizing information from an ensemble of snow profiles is an effective way to identify snowpack patterns in a forecast area. Identification and summarization tasks can be done fast and effectively by deriving visual summary statistics from distributed visual information. For example, humans can visually calculate correlation coefficients, clusters, and averages with their visual perception systems (Szafir et al., 2016). The volume and continuity of data produced by snowpack models offers new

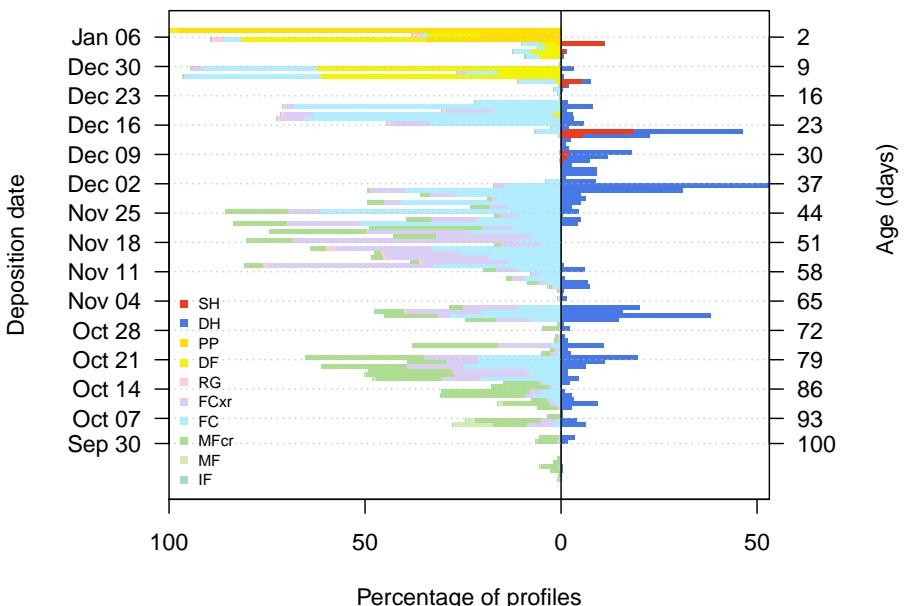

**Figure 5.** A visualization to identify avalanche problem types from 1180 simulated profiles on 8 January 2018. Snowpack layers are aggregated by their age to show their prevalence throughout the region (with widespread layers appearing in a greater percentage of profiles). A diverging scale distinguishes the layers with grain types associated with persistent weak layers (i.e. surface hoar and depth hoar) on the right from the layers containing other grain types on the left. Grain types are coloured using the perception-informed palette from Table 1.

opportunities for summarizing snowpack structure that are not possible with manual snow profiles. When used in combination with a colour palette that emphasizes snowpack features related to avalanche problems, profile ensemble visualizations can help forecasters identify prominent avalanche problem types.

A simple and powerful summary is obtained by plotting multiple grain type profiles side-by-side (Fig. 4). In this example, 1180 profiles are sorted from thinnest to thickest and over 46,000 individual snow layers are shown in a single view. Despite the large volume of data, a few prominent snowpack features pop-out and attention is drawn to the main snowpack patterns in the forecast area. Since this visualization is specifically designed for the task of identifying potential avalanche problem types, other idioms are required for visualizing geospatial patterns in a meaningful way (see Sect. 3.3).

Another summary visualization that draws attention to potential avalanche problem types is produced by aggregating layers by their age (Fig. 5). The simulated profiles for 8 January 2018 have layers dating back to the start of the winter. Common features amongst the set of profiles such as new snow near the surface and widespread weak layers share similar deposition dates, thus counting the number of profiles with different age and grain type combinations results in a summary of the snowpack structure. In Fig. 5, grain types associated with persistent weak layers are emphasized with a diverging horizontal scale to distinguish them from other grain types. The persistent weak layers are easier to notice in this visualization than in Fig. 3 and 4, because they occupy a greater spatial area in the visualization than visualizations where the spatial area occupied by a layer

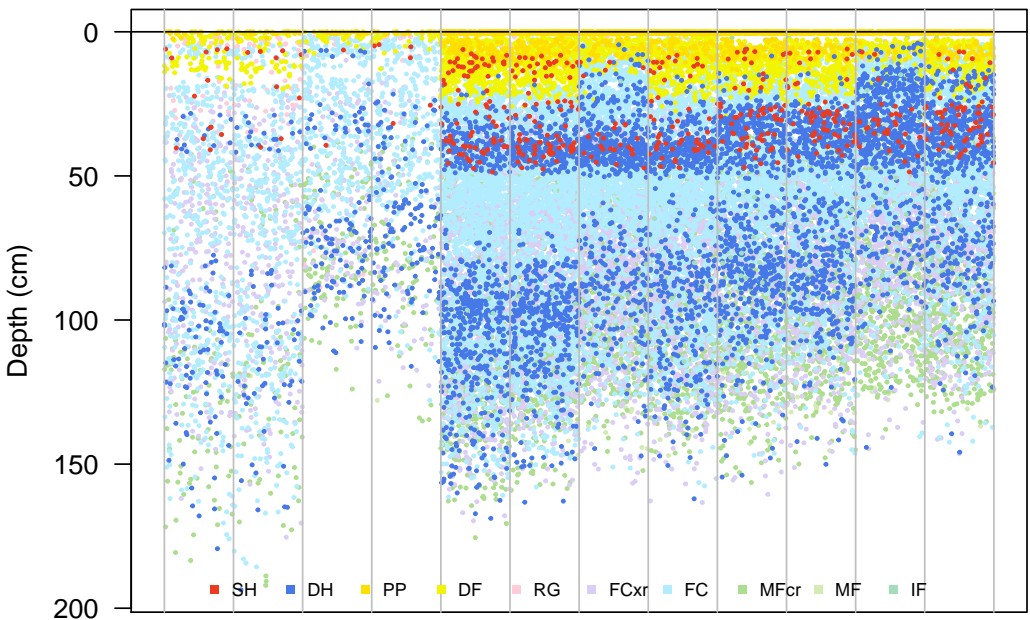

**Figure 6.** A visualization to locate avalanche problems in terrain. Snowpack layers from 1180 simulated profiles on 8 January 2018 are partitioned into terrain class bins for elevation band and aspect. Elevation bins include alpine (ALP), treeline (TL), and below treeline (BTL) and aspect bins include four cardinal directions (north, east, south, west). Each layer is given a random horizontal position within the bin to allow visual summary statistics. Grain types are coloured using the perception-informed palette from Table 1.

is proportional to its thickness. It is also possible to produce an aggregated stratigraphy profile by aligning layers based on other properties such as hardness (e.g. Hagenmuller and Pilloix, 2016), however this requires complex data transformations and assumptions about averaging layer properties. The layer prevalence visualization in Fig. 5 supports the task of identifying potential avalanche problem types in a way that is fast and simple to implement.

The visualizations in Fig. 4 and 5 use colour and position to draw attention to snowpack features that relate to the storm slab and persistent slab avalanche problems on 8 January 2018. The storm slab problem is apparent from the yellow new snow grains on the surface and a potential persistent slab avalanche problem is apparent from the salient surface hoar and depth hoar layers that are buried 30 to 50 cm below the surface (Fig. 4) and formed in early December 2017 (Fig. 5).

### 3.3 Locate avalanche problems in terrain

When locating avalanche problems in terrain, the description of the terrain depends on the context and scale of the forecast (Statham et al., 2018). For example, regional forecasters describe terrain by elevation bands and aspects while highway forecasters reference named avalanche paths. Partitioning snowpack data into distinct terrain classes and comparing side-by-

side views of the data for each terrain class is an effective way to visualize complex geospatial patterns. High-dimension (3D) visualizations are tempting to characterize mountainous terrain, particularly with high density model datasets, but there is large potential for misinterpretation on two-dimensional displays due to depth perception issues and over-plotting (Ware, 2012). Instead, simultaneously comparing one- or two-dimensional visualizations for different types of terrain has low cognitive load and less potential for misinterpretation.

To provide insight into the location of avalanche problems, the simulated profiles from Glacier National Park were partitioned into bins for elevation band and aspect classes to support regional-scale forecasting (Fig. 6). Avalanche forecasters often use radial plots to visualize simple aspect and elevation patterns such as danger ratings or the presence of an avalanche problem. While radial plots are familiar and widely used because of their metaphor for cardinal directions, the skewed and unaligned coordinate plane makes precise comparisons much more difficult. Given the complexity of snowpack model data, Fig. 6 uses rectilinear plots with an aligned scale for more accurate comparisons (Cleveland and McGill, 1984). A randomized horizontal position (i.e. jitter) was applied to each layer to reduce over-plotting and randomize the order within a bin (Ellis and Dix, 2007). The jitter plot allows the user to derive visual summary statistics about the snowpack structure in each terrain class and make comparisons between different terrain bins such as:

- snow depth generally increases with elevation, except for south and west facing slopes in the alpine,

- there is more new snow on north and east aspects,

- buried surface hoar layers are more prevalent on north and east aspects, and

- the early December 2017 weak layer is more prevalent at treeline and below treeline elevations.

These types of visual patterns could help forecasters localize avalanche problems in their terrain. Different types of terrain bins could be applied for other forecasting contexts to highlight differences between relevant types of terrain such as sub-regions, avalanche paths, or classes of ski terrain (e.g. Sterchi et al., 2019).

## 3.4 Compare distributions of avalanche size and likelihood

Avalanche size is easily visualized by aligning layers by depth rather than height. Layer depth is more relevant to forecasting avalanches than layer height, as weak layer depths correlate to the destructive potential of slab avalanches (McClung, 2009). From an information visualization perspective, comparisons are more effective on aligned scales, and thus aligning layers by depth allows users to browse the distribution of depths for specific weak layers. From the distribution of layer depths in Fig. 4 and Fig. 6, forecasters could estimate the potential sizes of storm slab and persistent slab avalanches. The distribution of layer depths in these visualizations relates to spatial variability amongst the profile locations. Overlaying summary statistics on the visualizations, such as the median depth of a specific layer, could further help estimating the size of avalanches in different types of terrain (as done in the interactive dashboard in Sect. 3.5).

The CMAH defines the likelihood of avalanches as a combination of sensitivity to triggers and spatial distribution (Statham et al., 2018), making it a relatively difficult attribute to visualize. Options for visualizing avalanche likelihood could include

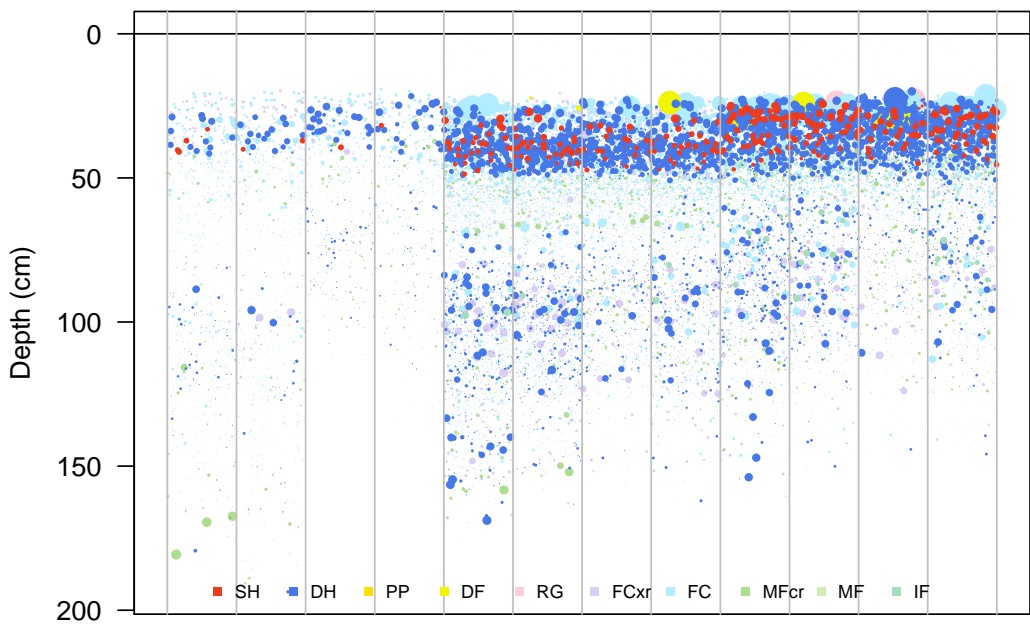

**Figure 7.** Visualization designed to show the likelihood of persistent slab avalanches by combining spatial distribution and the sensitivity to triggers of snowpack layers. Snowpack layers from 1180 simulated profiles on 8 January 2018 are partitioned into terrain class bins for elevation band and aspect. Elevation bins include alpine (ALP), treeline (TL), and below treeline (BTL) and aspect bins include four cardinal directions (north, east, south, west). The number of dots with persistent grain types in a terrain bin relates to the spatial density of the problem and the size of each layer's dot relates to its sensitivity to triggers (derived from the structural stability index). Each layer is given a random horizontal position within the bin to allow visual summary statistics. Grain types are coloured using the perception-informed palette from Table 1.

encoding related attributes with visual features such as shape, size, or motion in any of the previous idioms or by designing new idioms that focus specifically on likelihood. We present examples of both approaches using some simple attributes related to sensitivity to triggers and spatial distribution.

When working with snow profiles, one potential method for assessing the spatial distribution of a problem is counting relevant layers amongst a set of profiles as an indication of spatial density. Meanwhile, sensitivity to triggers can potentially be assessed with snowpack tests, stability indices, or structural criteria such as grain size and hardness (Schweizer and Jamieson, 2007). Snowpack models offer several stability indexes based on the mechanical and structural properties of the layers (Schweizer et al., 2006). We derive a relative measure of sensitivity to triggers ($S$) from SNOWPACK's structural stability index ($SSI$). The $SSI$ combines a stress-strength ratio with differences in hardness and grain size to calculate a value between 0 and 6, where lower values correspond to less stable layers. The $SSI$ is most effective for avalanche problems

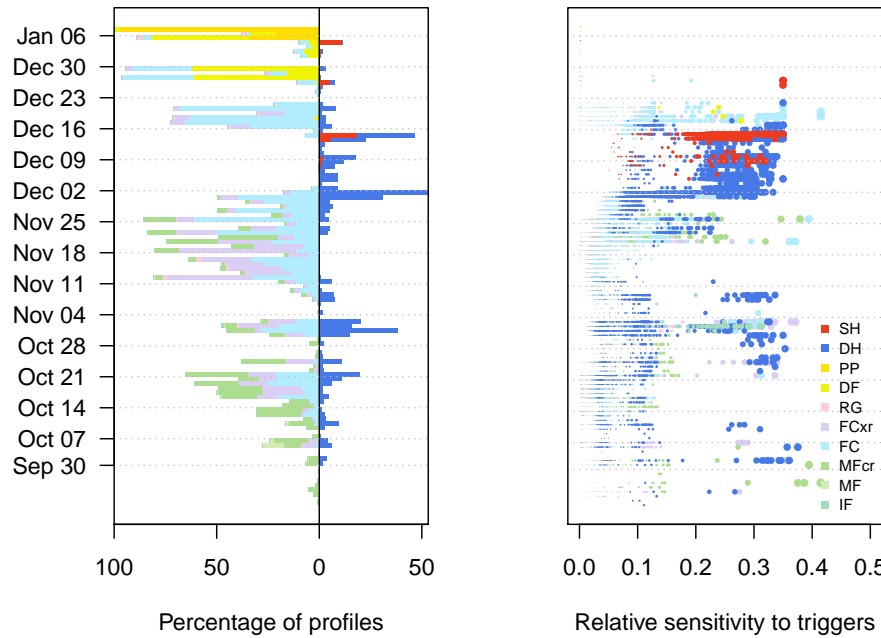

**Figure 8.** Combining visualizations of spatial distribution and sensitivity to triggers to provide information about the likelihood of avalanches from 1180 simulated profiles on 8 January 2018. Both visualizations aggregate the layers by age and colour them by grain type. The left panel shows spatial distribution by counting the number of profiles with different grain types (see Fig. 5) and the right panel shows the distribution of sensitivity to triggers for these same layers as derived from the structural stability index (with dot size proportional to sensitivity to triggers). Grain types are coloured using the perception-informed palette from Table 1.

associated with deep weak layers (e.g. persistent slab problems), because it ignores surface layers within skier penetration depth. To visually emphasize unstable layers, $SSI$ was transformed into a relative measure of sensitivity to triggers:

$$S \propto exp^{-SSI} \tag{1}$$

where the $SSI$ for each layer is scaled inverse exponentially to produce an ordered variable that correlates with the sensitivity categories from the CMAH (i.e. unreactive, stubborn, reactive, touchy). This transformation produces values between 0 and 1 and exaggerates differences for unstable weak layers with low $SSI$. The numeric value of the sensitivity measure does not have an interpretable meaning but can illustrate relative patterns when applied in visualizations.

We present two examples of visualizing likelihood information with this relative measure for sensitivity to triggers. The terrain class visualization in Fig. 6 was modified to scale the dot size of each layer to its sensitivity to triggers (Fig. 7). This creates greater emphasis on sensitive weak layers, so the combination of the number and size of weak layer dots in a terrain bin relate to the likelihood of persistent slab avalanches in that type of terrain. Another visualization specifically designed for likelihood is given in Fig. 8, where the left panel provides information about the spatial distribution of each layer and the right panel provides information about their sensitivity to triggers. Information about the spatial distribution is shown with the same visualization as Fig. 5, where the prevalence of each layer by age is related to the spatial density of the problem. Sensitivity to

triggers is shown with the distribution of the relative sensitivity of each layer by age. The side-by-side comparison of spatial distribution and sensitivity to triggers provides information about the potential likelihood of persistent slab avalanche problems. For example, the weak layers that formed in early December 2017 are more widely distributed and sensitive to triggers than the weak layers that formed in late October (i.e. avalanches are more likely).

It is important to note that we are presenting these likelihood visualizations to illustrate the concept of visually encoding stability information rather than suggest these derivations for an operational tool. These derivations are most effective for persistent and deep persistent slab avalanche problems, while the likelihood of other avalanche problem types may be better represented by other attributes such as weather variables or snow temperatures (Haegeli et al., 2010). Deriving stability information from simulated snow profiles is an active research topic (Monti et al., 2014), and new stability indices will likely provide more accurate information about the likelihood of avalanches.

## 3.5 Interactive dashboard

The visualizations presented in this section were combined into an interactive dashboard using *Tableau* data visualization software (Fig. 9). The dashboard facilitates the sequential questions of the CMAH by following the "overview first, zoom and filter, details on demand" mantra (Shneiderman, 1996). Interactions that allow the user to change the view by selecting visual features and filters from the legend. The initial view (Fig. 9a) consists of the layer prevalence visualization from Fig. 5, the profile summary visualization from Fig. 4, and the location in terrain visualization from Fig. 6. The combination of these visualizations provides a visual overview of the snowpack structure to support the first question in the CMAH – identifying potential avalanche problem types. After identifying potential avalanche problem types from the overview visualizations, users select layers of concern from the layer prevalence panel to update the visualizations. Once a layer of concern is selected, the layer is highlighted in the other panels to provide details about the location in terrain and the distribution of avalanche sizes (Fig. 9b). Horizontal bars show the median depth of the selected layer in each terrain class for comparison of potential avalanche sizes. A tooltip allows the user to hover over any visual feature and see details such as the grain type, deposition date, and depth in a pop-up window. In Fig. 9b, the user has selected all the layers that formed between 2 and 15 December 2017 to investigate the persistent slab avalanche problem. The profile summary shows the position of this layer in the snowpack and the location in terrain visualization shows the layer is more prevalent at treeline and below treeline, with median depths of $40\,\mathrm{cm}$ at treeline and $35\,\mathrm{cm}$ below treeline. Appendix A provides examples of this dashboard for several days throughout the 2017-18 winter.

## 4 Implementation

### 4.1 Design considerations

The visualizations presented in Sect. 3 are a starting point of how information from snowpack models can be designed to address specific forecasting needs, but additional user testing is necessary for them to evolve into a valuable forecasting tool.

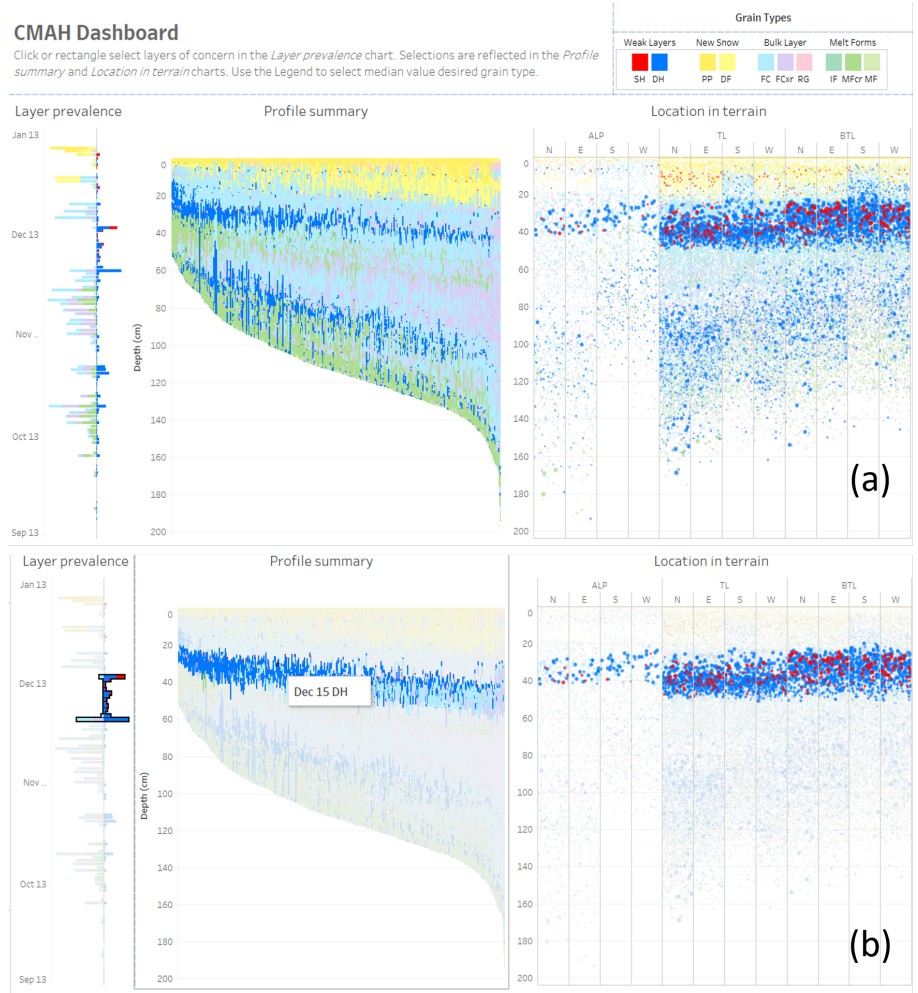

**Figure 9.** Screenshots of an interactive dashboard that provides visualizations of layer prevalence, profile summary, and location in terrain from 1180 simulated profiles on 8 January 2018. The initial view (a) provides and overview of the entire dataset for the user to assess potential avalanche problems and then (b) the updated view after the user has selected layers that formed between 2 and 15 December 2017 to explore details about the distribution and depth of the persistent slab avalanche problem.

The nested model for visualization design of Munzner (2009) provides a structured approach to evaluating the design of such tools, where issues can be addressed at each specific design level (i.e. domain situation, task and data abstraction, visualization and interaction idiom, algorithm).

At the domain situation level, creating links between snowpack models and avalanche forecasting workflows like the CMAH will address operational challenges faced by forecasters. Reflecting the broad adoption of the CMAH (Statham et al., 2018), the proposition of using snowpack models to characterize avalanche problems across forecast regions has gained more interest from the Canadian forecasting community than previous snowpack model tools that focused on individual stratigraphy profiles. However, the CMAH may not characterize all possible domain situations for snowpack models, as tasks such as terrain selection or civil protection possibly require distinct domain level considerations.

At the task and data abstraction level, designs should focus on specific questions and forecasting tasks. This requires a shift from bottom-up scientific visualizations towards top-down information visualizations. The visualizations in Sect. 3 are specifically designed to answer the four questions posed by the CMAH by focusing on the type of task (e.g. identify, locate, compare). Forecasters have existing methods for performing these tasks with field data, but aggregating and summarizing that data can be challenging and uncertain. The continuous spatial coverage of snowpack models offers unique opportunities to support these tasks. The side-by-side profiles (Fig. 4) and terrain class plots (Fig. 6) visualize snowpack patterns in ways that are not possible with traditional snow profile data and can help forecasters build a more complete mental model of the snowpack structure in their forecast area.

While the examples in Sect. 3 are particularly suited to storm and persistent slab avalanche problem types, the same principles could be applied to emphasize attributes important to other problem types (such as weather data to identify wind slab avalanche problems and snow temperature data to identify wet avalanche problem types). The task of locating avalanche problems in terrain differs for different forecasting contexts. In many cases maps or other geospatial visualizations could be valuable for this task. While not directly a question in the CMAH, the task of tracking temporal trends is also critical, as the forecasting process is iterative throughout the winter. The continuous temporal data provided by snowpack models offers unique capabilities for tracking snowpack evolution. Stratigraphy timeline visualizations (e.g. Fig. 3) are well suited for tracking snowpack evolution at individual locations, however adding a temporal dimension to spatial information creates additional complexity and requires specific design considerations. A visualization showing the temporal evolution of a snowpack summary would be particularly interesting. Examples of temporal change are showing in Appendix A with examples of the interactive dashboard for six different periods over the 2017-18 winter.

At the visualization and interaction idiom level, following established perceptual and cognitive principles ensures designs are effective at their intended tasks. The perception-informed colour palettes (Table 1 and 2) are examples of applying these principles to draw attention to the features in snow profiles that are deemed most important. The standard grain colour palette may emphasize features without intending to do so and is likely ineffective for individuals with colour blindness. The simplified colour palette in Table 2 could potentially be more relevant for forecasters as it shifts the purpose of colour from showing snowpack structure towards identifying avalanche problems. Considering information visualization principles listed in Sect. 2.4 could prevent data from being misinterpreted.

At the algorithm level, interactive tools need to be efficient in terms of time and memory performance. We tested several versions of interactive dashboards with operational forecasters. While these dashboards were not optimized for web performance, they worked at reasonable speeds with maximum wait times of 2-3 seconds for filtering layers in large regions with over 5000 profiles. Over-plotting becomes an issue for large data sets where the total number of layers becomes larger than the number of pixels on the screen, but can be addressed by stratified sampling or implementing subpixel rendering techniques to increase the apparent resolution of the screen. The visualizations may also be less effective with small data sets where there are not enough layers for patterns to emerge. This could be addressed by downscaling the model to increase the number of modelled profiles. An important consideration in data set size is ensuring the model is configured to capture an appropriate amount of spatial variability across the forecast region.

## 4.2   Steps towards operational implementation

While these designs in Sect. 3 are informed with well established visualization principles, user testing is critical to validate their actual operational value. Various versions of the interactive dashboard presented in Sect. 3.5 have been tested with operational forecasters in Canada, resulting in an agile development process where qualitative feedback has provided new perspectives and identified issues with the designs. An iterative process of feedback and redesign is critical for successful implementation of new visualization tools into operational workflow and is much less risk-prone than developing visualization tools in entirety. For example, the US National Weather Service used an agile development process to deploy their modern forecasting tool over several years in the early 2000s (LeFebvre et al., 2003).

The visualization design principles presented for snowpack model data are equally relevant for visualizing traditional field data. An ideal implementation of snowpack models into forecasting workflows would be combining field data and model data into a single interactive tool. A major motivation for adding model data into forecasting workflows is to reduce uncertainty about snowpack conditions. A visualization tool with mixed data sources would allow forecasters to assess the integrity of the model output as well as place the field observations into a broader context with the continuous spatial coverage of models. Many of the visualizations presented in Sect. 3 could be modified for such comparison tasks. Similarly, visualizing an ensemble of model data sets (e.g. with different meteorological inputs or geometric configurations) would provide insights about the confidence in modelled data.

In addition to improved visualization, model development and validation remains critical to improving the integrity of model output. This should continue in parallel to user testing so forecasters can offer operational feedback on model accuracy. Assimilating field data into snowpack models could greatly improve their integrity (Winstral et al., 2018), however model developers are faced with assimilation challenges such as mismatched spatial scales between gridded models and point field observations. Interactive visualizations of heterogenous field and model data has potential for researchers and forecasters to gain a deeper understanding of how they relate, and the knowledge gained through such a process can translate to improved computational assimilation methods.

# 5 Conclusions

We present visualization design principles that increase the interpretability and relevance of snowpack model outputs. These are two of the four major perceived issues with operational snowpack model tools identified by Morin et al. (2020) (besides accessibility and integrity). The nested model for visualization design (Munzner, 2009) provides a framework for defining the domain of avalanche forecasting and the necessary tasks that are needed to analyze data. Tasks required to assess avalanche hazard are described by applying the widely adopted conceptual model of avalanche hazard (Statham et al., 2018). From these tasks, we show how information visualization principles can be applied to design visual representations of snowpack model data in ways that leverage the human visual system to understand the complex nature of the data. A key idea in these designs is shifting from bottom-up scientific visualizations towards information visualizations that address user needs.

A critical next step is implementing and testing these designs in operational forecasting workflows. By addressing issues with the interpretability and relevance of snowpack model data, these designs will allow forecasters to learn the capabilities and deficiencies of snowpack models in a meaningful way. The same design principles should be considered when visualizing other types of avalanche and snowpack data, as the same domain situation and task abstractions apply when forecasters analyze field observations. Interaction idioms should play an important role in understanding complex model data, as they allow users to perform custom queries, test and validate hypotheses, and discover inconsistencies and anomalies. Interactions that compare model data with observation data would be particularly powerful in building trust in the models and addressing issues with their integrity. This process was critical in the adoption and trust in numeric weather predictions models by meteorologists (Benjamin et al., 2019), and just like meteorologists, avalanche forecasters could become active participants in model validation and improvement.

## Appendix A: Visualizations for different snowpack conditions

.  The code and data used to produce the visualizations are publicly available at https://osf.io/g5r7k (Horton et al., 2020). The interactive dashboard is available at https://avalancheresearch.ca/pubs/2020_horton_snowpackvis.

.  All authors worked on the conceptualization of this paper. SH prepared the data and software, SN contributed to visualization ideas and designs, and PH provided supervision. SH prepared the manuscript with review and editing from the other authors.

.  The authors declare no competing interests.

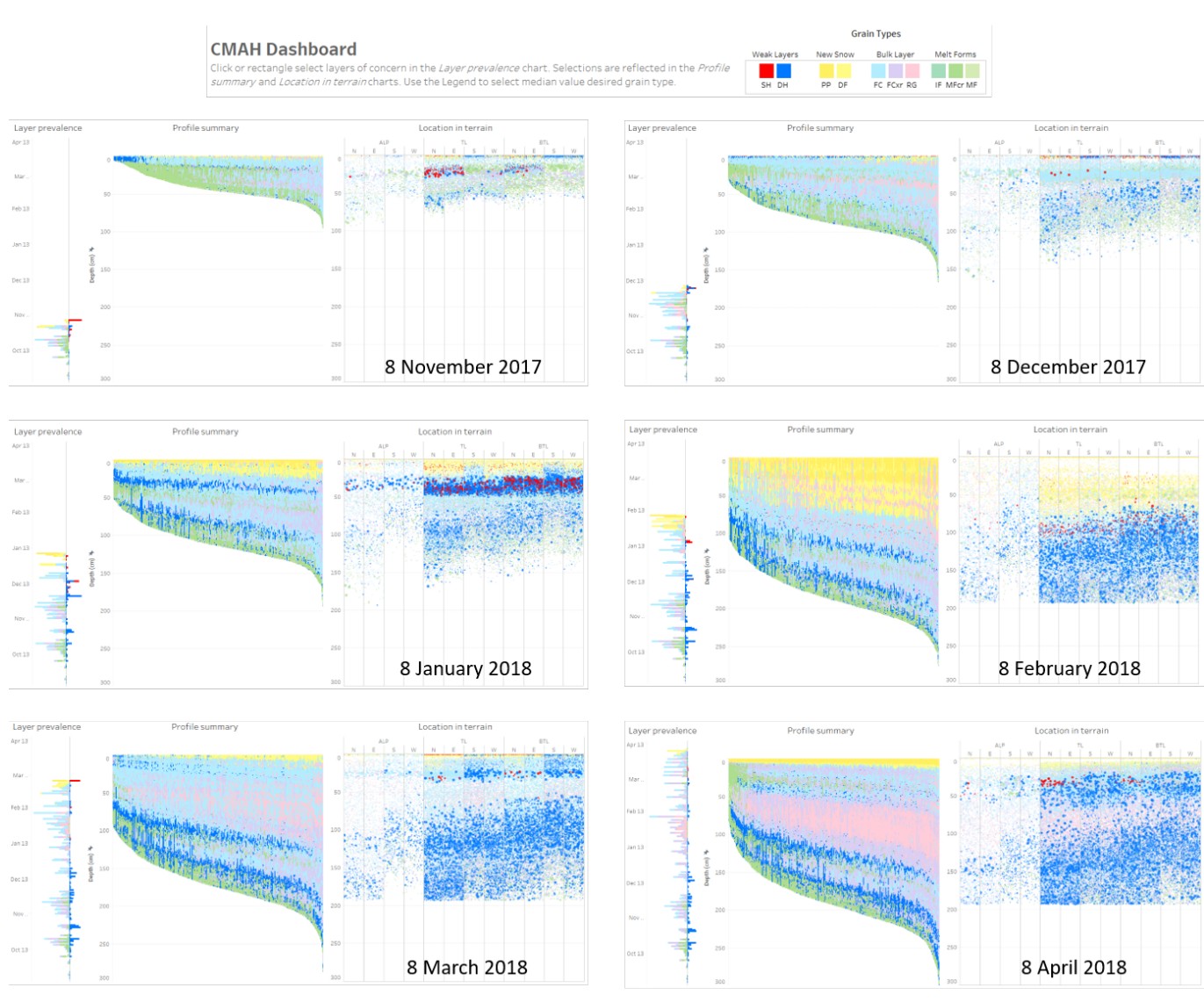

**Figure A1.** Comparison of the interactive dashboard for different days over the course of the 2017-18 winter. Each dashboard includes visualizations of layer prevalence, a profile summary, and location in terrain for the same 1180 simulated profiles in Glacier National Park.

. Thanks to Lyn Bartram of the Vancouver Institute of Visual Analytics, the Big Data Hub at Simon Fraser University (SFU), and the rest of our colleagues at the SFU Avalanche Research Program. The NSERC Industrial Research Chair in Avalanche Risk Management at SFU is financially supported by Canadian Pacific Railway, Helicat Canada, the Canadian Avalanche Association, and Mike Wiegele Helicopter Skiing. The SFU Avalanche Research Program is further supported by Avalanche Canada and the Avalanche Canada Foundation. We thank Karsten Müller, Jan-Thomas Fischer, and Michael Warscher for their constructive reviews.

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
