# Peer review of "Enhancing the operational value of snowpack models with visualization design principles"

_Natural Hazards and Earth System Sciences, 2019_

## Referee Comment (RC1) · Karsten Müller (Referee) · 2 Jan 2020

—SUMMARY—

The manuscript applies the visualization design framework proposed by Munzner (2009) to an established workflow for avalanche hazard assessment (CMAH). The goal is to enhance the interpretability and increase the relevance of numerical snowpack models for the avalanche forecaster. Snowpack models in avalanche forecasting are the equivalent to numerical weather prediction models (NWP) in weather forecasting. While weather forecasting nowadays heavily relies on NWPs, snowpack models are only sparsely used in operational avalanche forecasting. Accessibility, interpretability,

relevance and integrity of snowpack models are not yet good enough for operational purposes a recent study by Morin et al claims. While issues of accessibility and integrity are not addressed in this manuscript, the main reason for poor interpretability is accorded to poor visualization of snowpack model output, which also reduces their relevance to the avalanche forecaster. Existing visualization tools are designed by the model developers with evaluation of model performance in mind, but not the operational forecaster (end-user). This manuscript presents a top-down approach for visualizing snowpack models with the forecaster/user in mind. The authors suggest that the visualization of snowpack model output should help the forecaster to answer the key questions from the conceptual model of avalanche hazard (CMAH). They demonstrate their design by applying it to a regional avalanche forecasting scenario. The study concludes that the presented top-down design approach is superior in an operational setting based on a small user survey.

—GENERAL COMMENTS—

This work is a relevant technical contribution. It describes a thorough process on how to improve snowpack visualization in operational avalanche forecasting. The avalanche community in North America, but also internationally, will benefit from these results. I found it well written and structured. Figures are of high quality. I recommend to publish the manuscript. I only have minor suggestions on how to improve the paper.

It seems like the target audience for the suggested visualizations are regional avalanche forecasters that operate with forecasting areas of several hundreds to thousands sq.km. The language used and the given example address this audience. I miss a discussion on other operational settings and more extreme cases, i.e. very large forecasting areas and high resolution model (large amount of data) and small forecasting regions and poor model resolution (too little data). Could you discuss what needs to be done to transfer the presented plots to a smaller scale? Can these visualizations be beneficial for managing avalanche safety in a ski resort? Is there a minimum number of simulated snow profiles to apply your designs? On the other hand, an avalanche

forecaster can normally not wait for several minutes for a plot to be displayed. Could you discuss performance on a standard workstation/PC? How long does it take to load and update the dashboard in your example on common hardware?

In my opinion an important part of showing model output is to provide a measure/display of uncertainty or an indication of when the model is off. In my experience, forecasters rejected model output because it is often hard and time consuming to evaluate if the model is providing reasonable results. Thus, be able to plot model output against field data or other sources will help to improve integrity. You mention this at the end of your conclusion. Could you, in addition, discuss briefly why this is not part of your study? I suggest to add a short section that summarizes the main issues with assimilating snowpack observations in snowpack models.

A drawback of this study is the small and rather arbitrary user survey on the effectiveness of the proposed visualization design.

—TECHNICAL COMMENTS—

p2 l8: should be "Morin et al. (in press)" as in prev sentence

p2 l10: What is meant by "workstations"? To me this is hardware - a PC! They are normally not specifically designed for showing field data. I assume you address the lack of proper software that can make model output accessible to the forecasters. A hardware issue might be lack of CPU/GPU power to effectively handle large amounts of data/images.

p2 l13: Snowpack models "relevance" comes from their ability to produce information over a large spatial scale, something field observations can not. So I think their relevance is less of a problem than their integrity, i.e. difficult to compare to field observations due to scale issues in the forecing data.

p2 l17: Could you provide an example of a "conventional method" - individual snow profiles?

p3 l19: You could provide "assess the spatial distribution of weak layers" as an example for "major needs".

Figure 1: The figure caption could be more detailed and explain the main features and abbreviations of the shown chart. This would ease the understanding for readers outside the avalanche community.

p7 l18-19: The given example is typical for a regional avalanche forecast and the presented visualizations work well in this given case. Could you discuss (later in the text) the extremes, very large forecasting areas and high resolution model (large amount of data) and small forecasting regions and poor model resolution (too little data).

Table 2: Layers of large facets can be an avalanche problem, too. However, facets (FC) are treated as bulk layers here. What is the criteria (in SNOWPACK) that separates DH from FC - size only - if yes, what is the threshold?

p9 l5: remove "and"

p9 l11: "...each day OF the season."

Figure 4: Can you explain why the percentage exceeds 100% on some days? E.g. Oct 21 or Noc 22 and 25. Does the plot only evaluate if a layer is present (boolean - regardless of layer thickness) in a simulated profile or is the percentage each layer takes up of the total snow depth within each simulation regarded and used as a form of weight?

Figure 5: Have you considered a "polar/radar plot" for each elevation band? I can imagine that it will show the presence of layers with regard to aspect more clearly. However, the information on snow depth will be difficult to integrate.

p16 l 2: "...the principles outlineD in..."

p16 l 16: Could you provide examples for "other operational tasks"? And discuss briefly benefits and challenges (see General Comments).

p16 l26-27: What do you mean by "...prototypes were accessed externally from existing workstation..."? Were the plots provided by a server? Why is it critical to integrate visualizations into forecasting workstations? Do you mean integrate into software used by the forecasters?

p17 l3: Consider to add "...two of the four major percieved issues (BESIDES ACCESSIBILITY AND INTEGRITY) with ..."

p17 l11-12: The sentence "As highlighted by..." should be removed or rephrased since it is not clear how it fits into your conclusions. Please be more specific on your findings.

---

## Referee Comment (RC2) · Jan-Thomas Fischer (Referee) · 5 Feb 2020

In this manuscript the authors present a new methodological approach to enhance the operational information value of snow pack models. One of the main developments (that could get even more attention) is the (easy to use) CMAH dashboard tool, providing an online, interactive approach to reproduce the main outcomes/figures of the paper. The authors successfully demonstrate how their approach enhances the information value of snow pack modelling, with particular emphasis of layer prevalence and their spatial distribution. The paper also shows that future research and work is necessary to implement an appropriate measure and visualization for stability. The

authors succeed in highlighting the spatially distributed character of the results (which could deserve an additional *spatial* description, see comment below). Besides this spatial character it would be worth to elaborate (or at least mention) how this method can be applied with respect to temporal variability. This could e.g. be achieved by (1) evaluating/providing an additional CMAH dashboard for a different date (in mid December?) as supplementary material (if the corresponding workload allows to?) or by (2) briefly discussing how the content and information value changes/develops throughout a season (cf. seasonal variation in Fig 2).

All in all the paper is well written (although the authors could review/explain which and why so many terms are italicized). The paper has a good mixture of technical terms and corresponding descriptions. It is of high quality, enjoyable to read and fits to the scope of NHESS.

**Please find some detailed line-by-line comments/questions below:**

- *p1 l 8, ...in terrain.*: Please specify on which scale(s).

- *p2 l 1, ...can range from individual slopes in backcountry guiding.*: I do not see how individual slopes are relevant in the context of this work. Is it possible to distinguish?

- *p3 l1, ...so far prevented...:* ... so far limited...

- *p5 l5, ...is analogous to manual snow stratigraphy...*: This is very important and could be further highlighted throughout the paper by providing (a) manual profile(s) representative for the date / location(s) of the main analysis (e.g. 8 January 2018). Further the value of Figure 1 could be enhanced by connecting the snowpack information of the generalized profiles to the study area/time.

- *p7 l17-19:* Could you comment on the specs of the profile locations and how they are chosen/defined to be representative for the study region? Is the number of study plots important? What number is expected to provide a representative analysis for the region (or spatial density)? Since spatial distribution is an important point of your analysis it could be worth to provide a corresponding map overview of the study region and profile locations (in particular for readers that are not familiar with the region).

- *p7 l20-25:* Why is the emphasis undesired? Could you elaborate a bit with respect to what some features have high or low importance?

- *p8 l7-9, Table 2:* Could you please add some references and comments on the simplified groups of grain types, with particular emphasis on why it would or why it would not be appropriate to summarize RG and FC as bulk layers (with respect to different types of metamorphism / underlying physical process). In your example (see e.g. Fig 2) it could appear also appropriate to add MF as bulk layer?

- *p9 l11-12:* Please Specify (see also comment on Figure 4). Are you displaying the percentage for a specific date (8 January 2018)?

- *p11 l14:* Could you provide a (technical) reference for the "jitter" plot?

- *p12 l10-14*: I think it would be worth to (1) mention the availability of the median values in the interactive dashboard (which are way more instructive than the figure) and (2) to comment on the spatial variability of throughout your profiles, e.g. by mentioning the standard deviation and median values for the expected depth main avalanche problems.

- *p13 l6:* Is prevalence really connected to spatial distribution (including all sample pits) or is a total measure of occurrence?

[Figure]

- *p13 l9-10:* I think it would be worth to shortly discuss why the storm slab (that previously appeared as one of the main problems) is not highlighted in the sensitivity to triggers?

- *p14 sec. 3.5:* I find this section highly instructive to understand figures and conclusions in this paper and would like to see interactive CMAH dashboard mentioned earlier in the paper, since it e.g. provides more instructive/clear visualization than the printed terrain class visualization (Fig 5).

The figures generally appear clear but would benefit from a (more) self-sufficient description (by e.g. referring to the interactive dashboard where applicable and indicating the specific date in all plots/captions (where applicable, e.g. of Fig 2 (right) and Fig 3, 4, 5, 6, 7, 8 )):

- *Figure 1:* Could you enhance this Figure / increase readability and describe which information is given in the generalized snow profile or alternatively provide a manual profile of the study region/time as reference (additionally a map view of modelled study plots could be beneficiary here, see comment below)?

- *Figure 2:* Please indicate/describe somewhere (text and caption) what the difference between the left (timeline of stratigraphy) and right (hardness vs depth for a specific date (which?)) plot are?

- *Figure 3:* It would be helpful somehow give an overview of the study area and where the profiles are simulated.

- *Figure 4, Figure 7:* How can 100% be exceeded (e.g. Nov 24/23 and Oct 22)? Please double check your scale or explain (see comment above concerning prevalence). In the caption - persistent grain types or grain types associated to potential/persistent weak layers (e.g. these are mentioned as weak layer in the dashboard, please double check for the sake of consistency)?

- *Figure 7: ...shows spatial distribution...* Is it really the spatial distribution in this case (like e.g.Fig 5), or rather a total measure of occurrence (see comment above). Would it be beneficial/feasible to also use the size scaling of Fig 6 - allowing for a visual comparison/connection btw. the Figures?

- *Figure 8:* Why are IF not specified/displayed in the dashboard?

---

## Referee Comment (RC3) · Michael Warscher (Referee) · 25 Feb 2020

In their manuscript "Enhancing the operational value of snowpack models with visualization design principles", the authors present the application of different visualization design principles in the domain of avalanche forecasting using data from the widely used model SNOWPACK.

General Comments

The manuscript is technically very well written, as well as easily readable and understandable. I list some general comments and specific remarks in the following.

I fully understand and appreciate the usefulness of the presented visualizations and their simplificatiion and aggregation character, however, I still would like to additionally see some conventional map plots at the top-level of the dashboard. This would be very helpful to get an overview of the domain and the spatial distribution of specific snow characteristics and avalanche problems. Examples of such visualizations are presented in e.g. Morin et al. 2020. Your aggregated plots would be a perfect summarizing and aggregating approach in a second visualization step.

In my opinion, the most important missing approach in the presented framework is the implementation of validation data. You state in different parts of the manuscript that practitioners lack trust in the integrity of model data. They won't gain any if they do not see the model performance at some validation points at a glance in the operational setup or at least in some hindcast simulations. I think some of the presented visualizations are perfectly suited to include observed validation data. You could simply include an interface to integrate measured snow profiles and plot them right into your visualizations as single highlighted data points or in the best case, somehow link them to their respective model grid point (this way, they could be included in all your visualizations, even the "sorted-by-depth" ones). I understand that it could be complicated to do this in a visually attractive way, but I think it would be well worth the effort.

While I very much like the presentation of your new color profiles, I am kind of torn as they are very much tailored to previous existing expert knowledge (potential weak layer = surface/depth hoar = highly visible) and is not very generic. Of course, this is very useful to detect the targeted wind slab avalanche problems, but what about other common avalanche problems (e.g. wet-snow avalanches). Are they also clearly visible in your visualizations? Regarding this remark, - if feasible - it would be very beneficial for the manuscript to include an additional example for a very different avalanche situation in the same domain.

I don't see the point of having so many words printed in italic letters even if they refer to specific technical terms. I think this is not necessary here and they could all just be

changed to normal fonts.

As the manuscript provides a technical report of the application of a visualization concept, it would be very beneficial to add information about the minimum requirements for a snowpack model in terms of resolution, simulation variables and output that is needed to feed the visualization software and dashboard. It is obvious that the software was developed for the use with SNOWPACK as a well-known and established snow (layer) model, but it would be interesting to read some more technical details about input requirements and portability.

It would also be useful to include some more variables displayed in your visualizations, e.g. depth profiles of snow temperatures or snow density which might also be useful for avalanche practitioners and should be provided by the SNOWPACK model.

The user survey presented in section 4 is very little explained and far from being representative, so you should consider removing the section and just move the last sentence of the section to your conclusions.

I have two other comments, which might well be beyond the scope of this paper, but could be a useful addition for the future development of the presented approach: In addition to the above-mentioned validation data, it would be very useful to provide a framework for ensemble simulations including uncertainty measures. The implementation of visualizations for multi-model results and corresponding model spreads and uncertainties (ensemble model outputs from e.g. different initial conditions, different meteorological forcing data, and different snow pack models) would be a logical and highly valuable (or even necessary) next step for the application of snowpack models in real-world operational avalanche forecasting settings (similar to NWP). You should add this somewhere in your conclusions. Another helpful addition for avalanche forecasters and practitioners would be the visualization of the meteorological input in your visualization framework, e.g. wind speed and gusts, (min./max./mean.) air temperature, liquid/solid precipitation, SW/LW radiation, all separated for elevation and aspect

bands and sectors (of course depending on resolution and origin of the gridded mete-
orological forcing, domain size, etc.).

Specific Comments

P. 1, L. 8/9: Rephrase the sentence "Examples of visualizations that support these
tasks are presented and follow established perceptual and cognitive principles from
the field of information visualization.", to e.g. "Examples of visualizations that support
these tasks and follow established perceptual and cognitive principles from the field of
information visualization are presented."

P.1, L. 18 and others: Regarding the term "workstations". Maybe Benjamin et al. 2019
labelled the development of software, more powerful computers and more available
model and observation data as kind of mythical "workstations", I would prefer just to
call it what it is, namely more powerful computers, more data, and better visualization
tools that gradually developed in NWP and of course in all other fields.

P. 2, L. 7 and others: update citation Morin et al., is published now.

P.5, L. 10: "as hardness profiles" instead of "as a hardness profiles"

P. 6. Fig. 1: Do you have a version with better image quality available? The figure is
very hard to read. However, I would suggest to remove Fig. 1 anyways as it does not
contain important information in the context of the manuscript. If you decide to keep it,
you should add some more information to the manuscript explaining what the reader is
supposed to see in the figure.

P. 9, Fig. 2, x-axes right panel-plots: Please add explanation for the hardness abbre-
viations and a "hardness" x-axis label. It becomes clear from the text, but should be
included in the figure or at least in the figure caption. That also holds for the hardness
test abbreviations (F, 4F, 1F, P, K) which are clear for an avalanche practitioner (fist, 4
fingers, 1 finger, pencil, knife), but the article might be interesting for a broader (snow)
scientific audience. Please add explanations.

[Figure]

P. 9, L. 15: "Herla et al., in preparation" should be removed if not already published by now.

P. 10, Fig. 3: Even if it is clear when reading the manuscript and figure caption, I would prefer to have an arrow-type label on the x-axis (e.g. "Thinnest snowpack <-> Thickest snowpack")

P. 11, L. 7: "way to visualize" instead of "way visualize"

P. 11, L. 10: I suggest to rephrase the sentence: "Instead, using eyes to...", e.g. "Instead, simultaneously comparing 1D/2D visualizations..."

P. 13, Fig. 6, caption: "slab" instead of "slabs"

P. 14, Fig. 7: Labels "Sep 30" and "Sep 23" overlap, please solve this issue.

P.14, L. 2: Please use italic here ("Tableau") as this seems to be the name of a commercial software developing company. Just a comment: it would be very beneficial if you would develop the dashboard in R or a similar open source programming language, as you have already done with the visualizations. This would foster the use of your very useful software by different target groups.

P. 15, Fig. 8: Could you provide a screenshot with better quality? The very useful dashboard is kind of hard to acknowledge here.

---

## Author Comment (AC1) · 2 Apr 2020

**GENRAL RESPONSE TO ALL REVIEWERS**

We thank Karsten Müller, Jan-Thomas Fischer, and Michael Warscher for their constructive reviews. We have revised our manuscript to address their concerns. We first respond to some of their common comments below, then address their remaining comments individually.

A common theme in the comments was requests for additional features in the visualizations. While we appreciate the suggestions, the intent of our paper was not to

design the ultimate visualization tool for avalanche forecasting, but rather to argue that snowpack models can be more useful by applying avalanche forecasting principles and design principles to develop user-focused tools. To address this potential miscommunication, we tried to strengthen the main messages in our revised manuscript and emphasize the reasons for the included visualization examples more clearly.

Another common theme was concerns over our limited user survey. We have removed the survey and instead simply discuss the validity of our designs. The design principles included in our paper are grounded in a large body of literature in visualization research and do not need to be explicitly validated. However, our revised manuscript clearly emphasizes the need for extensive user testing for the successful implementation of operational visualizations.

We also agree with the reviewers that incorporating validation data would add great value to these visualizations. We added more discussion about this being another critical next step, but choose not to include this in our visualization examples, as this is would be rather complex and could dilute the main message of our paper.

Finally, there were several comments about applying the visualizations to different forecasting contexts and snowpack conditions. We've added more discussion about different forecasting contexts and added an Appendix figure with the dashboard images for different snowpack conditions over the course of a season in our study area.

**RESPONSE TO RC1 (Karsten Müller)**

*The manuscript applies the visualization design framework proposed by Munzner (2009) to an established workflow for avalanche hazard assessment (CMAH). The goal is to enhance the interpretability and increase the relevance of numerical snowpack models for the avalanche forecaster. Snowpack models in avalanche forecasting are the equivalent to numerical weather prediction models (NWP) in weather forecasting. While weather forecasting nowadays heavily relies on NWPs, snowpack models are only sparsely used in operational avalanche forecasting. Accessibility, interpretability, relevance and integrity of snowpack models are not yet good enough for operational purposes a recent study by Morin et al claims. While issues of accessibility and integrity are not addressed in this manuscript, the main reason for poor interpretability is accorded to poor visualization of snowpack model output, which also reduces their relevance to the avalanche forecaster. Existing visualization tools are designed by the model developers with evaluation of model performance in mind, but not the operational forecaster (end-user). This manuscript presents a top-down approach for visualizing snowpack models with the forecaster/user in mind. The authors suggest that the visualization of snowpack model output should help the forecaster to answer the key questions from the conceptual model of avalanche hazard (CMAH). They demonstrate their design by applying it to a regional avalanche forecasting scenario. The study concludes that the presented top-down design approach is superior in an operational setting based on a small user survey.*

*GENERAL COMMENTS*

*This work is a relevant technical contribution. It describes a thorough process on how to improve snowpack visualization in operational avalanche forecasting. The avalanche community in North America, but also internationally, will benefit from these results. I found it well written and structured. Figures are of high quality. I recommend to publish the manuscript. I only have minor suggestions on how to improve the paper.*

*It seems like the target audience for the suggested visualizations are regional avalanche forecasters that operate with forecasting areas of several hundreds to thousands sq.km. The language used and the given example address this audience. I miss a discussion on other operational settings and more extreme cases, i.e. very large forecasting areas and high resolution model (large amount of data) and small forecasting regions and poor model resolution (too little data). Could you discuss what needs to be done to transfer the presented plots to a smaller scale? Can these visualizations be beneficial for managing avalanche safety in a ski resort? Is there a minimum number of simulated snow profiles to apply your designs? On the other hand, an avalanche forecaster can normally not wait for several minutes for a plot to be displayed. Could you discuss performance on a standard workstation/PC? How long does it take to load and update the dashboard in your example on common hardware?*

**Our focus on regional scale forecasting was in response to regional scale forecasters having the greatest interest in snowpack models so far. We have added more discussion about how the design principles could apply to other contexts by: at the start of Sect. 3 explaining the purpose of our examples are to show applications of design principles, acknowledging how different contexts have different spatial scales and descriptions of terrain that could benefit from maps or other geospatial views (other than elevation-aspect classes), recognizing the model needs to be configured to capture the variability of snowpack conditions for the type of forecast area (which requires model expertise). We also try to generalize more of our discussion about how the design principles could be applied to any forecasting application by emphasizing the need to focus on user needs and their specific tasks and questions. As far as technical specifications, we added a summary of the practical speed and data set size limitations of visualizations in Sect. 4.2.**

*In my opinion an important part of showing model output is to provide a measure/ display of uncertainty or an indication of when the model is off. In my experience,*

*forecasters rejected model output because it is often hard and time consuming to eval-uate if the model is providing reasonable results. Thus, be able to plot model output against field data or other sources will help to improve integrity. You mention this at the end of your conclusion. Could you, in addition, discuss briefly why this is not part of your study? I suggest to add a short section that summarizes the main issues with assimilating snowpack observations in snowpack models.*

We agree including field data to validate the models is a critical step towards fore-casters trusting models. We have expanded our discussion of this step in a new sec-tion (4.2 Steps towards operational implementation), and although field data could be added to some of our example visualizations, this is non-trivial and we think beyond the scope of the main message of our paper. We also discuss the how visualization could play a helpful roll in assimilating field data into models by allowing forecasters and researchers to explore relationships between the two data in rich ways.

*A drawback of this study is the small and rather arbitrary user survey on the effective-ness of the proposed visualization design.*

We agree the user survey was small. We are currently in the early stages of the feedback process with forecasters and do not have a full user analysis to report, so we removed reference to the survey efforts and instead described the importance of an iterative design process where qualitative user feedback plays a very important role in informing and improving designs. Instead of discussing the validity of our specific designs, Sect. 4.1 now focuses on recommendations for designs in various forecasting contexts at each level of the nested model of visualization design.

*SPECIFIC COMMENTS*

- *p2 l8: should be "Morin et al. (in press)" as in prev sentence*

  **Updated to fully published reference (Morin et al., 2020).**

- *p2 l10: What is meant by "workstations"? To me this is hardware - a PC! They*

*are normally not specifically designed for showing field data. I assume you address the lack of proper software that can make model output accessible to the forecasters. A hardware issue might be lack of CPU/GPU power to effectively handle large amounts of data/images.*

**Workstation was the term used in the Benjamin et al (2019) meteorology paper, but to be more specific we replaced that term with either "visualization tool" or "workflow" throughout the manuscript as appropriate.**

- *p2 l13:* Snowpack models "relevance" comes from their ability to produce information over a large spatial scale, something field observations can not. So I think their relevance is less of a problem than their integrity, i.e. difficult to compare to field observations due to scale issues in the forcing data.

  **We agree that snowpack models inherently should have 'relevant' information, but despite that, when Morin et al. (20209) discuss the information quality of snowpack models, under the 'relevance' section they suggest the added value of model information is unknown . The integrity is a serious issue, but the focus of our paper was relevance and interpretability. We have expanded our discussion on the importance of improving model integrity through visualization techniques.**

- *p2 l17: Could you provide an example of a "conventional method" - individual snow profiles?*

  **As suggested, we added "manual snow profiles" as an example.**

- *p3 l19: You could provide "assess the spatial distribution of weak layers" as an example for "major needs".*

  **Done.**

- *Figure 1: The figure caption could be more detailed and explain the main features*

*and abbreviations of the shown chart. This would ease the understanding for readers outside the avalanche community.*

**We have added a more detailed explanation of the information in this photo (and replaced with a high quality image).**

- *p7 l18-19: The given example is typical for a regional avalanche forecast and the presented visualizations work well in this given case. Could you discuss (later in the text) the extremes, very large forecasting areas and high resolution model (large amount of data) and small forecasting regions and poor model resolution (too little data).*

**These cases are now discussed in Sect. 4.1 under design considerations at the algorithm level.**

- *Table 2: Layers of large facets can be an avalanche problem, too. However, facets (FC) are treated as bulk layers here. What is the criteria (in SNOWPACK) that separates DH from FC - size only - if yes, what is the threshold?*

**This grouping is now explained and justified in more detail. In short, SNOW-PACK has a size threshold where FC larger than 1.5 mm are called DH. Thus most FC layers in the model appear in bulk layers while most problematic FC layers appear as DH. We also cite Schweizer and Jamieson (2007) for common rules about identifying weak layers (i.e. the combined importance of grain type and size).**

- *p9 l5: remove "and"*

**Done.**

- *p9 l11: "...each day OF the season."*

**Done.**

- *Figure 4: Can you explain why the percentage exceeds 100% on some days? E.g. Oct 21 or Noc 22 and 25. Does the plot only evaluate if a layer is present (Boolean - regardless of layer thickness) in a simulated profile or is the percentage each layer takes up of the total snow depth within each simulation regarded and used as a form of weight?*

  **We corrected the algorithm to avoid percentages over 100%. These were caused by cases when a single profile had multiple layers on the same date with different grain types, but to simplify the data we now choose a single layer per date (with priority for weak layers over new snow over bulk layers over melt forms). And yes, the plot only counts whether a layer is present and does not account for layer thickness. We have expanded our description of this method in the text to make it clearer.**

- *Figure 5: Have you considered a "polar/radar plot" for each elevation band? I can imagine that it will show the presence of layers with regard to aspect more clearly. However, the information on snow depth will be difficult to integrate.*

  **Yes, we've experimented with circular plots as these are a familiar idiom for weather and avalanche forecasting. Based on visualization literature we found circular plots are effective and intuitive for overview tasks for simple types of data, but precise comparisons are difficult due to the skewed and unaligned scales. So we added a sentence explaining this rationale and suggest circular plots for simple data (category, ordinal), but argue complex data is better suited to rectilinear plots, especially when the intent of the visualization is precise comparison tasks.**

- *p16 l 2: "...the principles outlineD in..."*

  **Sentence removed.**

- *p16 l 16: Could you provide examples for "other operational tasks"? And discuss briefly benefits and challenges (see General Comments).*

**This section has been rewritten and now focuses more broadly on how snowpack model data could be visualized for a broad selection of contexts and tasks.**

- *p16 l26-27: What do you mean by "...prototypes were accessed externally from existing workstation..."? Were the plots provided by a server? Why is it critical to integrate visualizations into forecasting workstations? Do you mean integrate into software used by the forecasters?*

  **We meant there was an accessibility barrier because the models were accessed on a separate webpages for the forecasters familiar workflows (software, bookmarks, etc.). However, we no longer discuss the specific user feedback.**

- *p17 l3: Consider to add "...two of the four major perceived issues (BESIDES ACCESSIBILITY AND INTEGRITY) with ..."*

  **Done.**

- *p17 l11-12: The sentence "As highlighted by..." should be removed or rephrased since it is not clear how it fits into your conclusions. Please be more specific on your findings.*

  **Sentence removed.**

**RESPONSE TO RC2 (Jan-Thomas Fischer)**

*In this manuscript the authors present a new methodological approach to enhance the operational information value of snowpack models. One of the main developments (that could get even more attention) is the (easy to use) CMAH dashboard tool, providing an online, interactive approach to reproduce the main outcomes/figures of the paper. The authors successfully demonstrate how their approach enhances the information value of snowpack modelling, with particular emphasis of layer prevalence and their spatial distribution. The paper also shows that future research and work is necessary to implement an appropriate measure and visualization for stability. The authors succeed in highlighting the spatially distributed character of the results (which could deserve an additional spatial description, see comment below). Besides this spatial character it would be worth to elaborate (or at least mention) how this method can be applied with respect to temporal variability. This could e.g. be achieved by (1) evaluating/providing an additional CMAH dashboard for a different date (in mid December?) as supplementary material (if the corresponding workload allows to?) or by (2) briefly discussing how the content and information value changes/develops throughout a season (cf. seasonal variation in Fig 2).*

**We agree that temporal trends are important for forecasting, and although now a specific question in the CMAH, the fact the forecasting process is iterative (and usually repeated daily) tracking temporal changes becomes important. The temporal continuity of snowpack models are well suited to understanding temporal trends, and we have added some discussion in Sect. 4.1 about how this is a specific task requires its own designs for this purpose (such as stratigraphy timelines). We added examples of the CMAH dashboard for several dates over the course of the season for Glacier National Park in the Appendix. This both shows potential to track temporal variability, and shows the effectiveness of the visualizations for different snowpack conditions.**

*All in all the paper is well written (although the authors could review/explain which and why so many terms are italicized). The paper has a good mixture of technical terms and corresponding descriptions. It is of high quality, enjoyable to read and fits to the scope of NHESS.*

**We removed most of the italics.**

- *p1 l 8, ...in terrain.: Please specify on which scale(s).*

  **We generalized the sentence and added "at relevant spatial scales."**

- *p2 l 1, ...can range from individual slopes in backcountry guiding.: I do not see how individual slopes are relevant in the context of this work. Is it possible to distinguish?*

  **This list is meant to show the diversity of avalanche forecasting contexts, we address how our visualizations and design principles transfer to other spatial scales in later sections.**

- *p3 l1, ...so far prevented...: ... so far limited..*

  **Done.**

- *p5 l5, ...is analogous to manual snow stratigraphy...: This is very important and could be further highlighted throughout the paper by providing (a) manual profile(s) representative for the date / location(s) of the main analysis (e.g. 8 January 2018). Further the value of Figure 1 could be enhanced by connecting the snowpack information of the generalized profiles to the study area/time.*

  **Yes, many of the ideas behind our designs are inspired from existing methods with manual profiles. We have added some more discussion of this, for example explaining how some of methods for grouping weak layers, characterizing sensitivity to triggers, etc. draw from established methods for interpreting manual profiles.**

We appreciate the value of having manual or whiteboard profiles for the same study area and period, however this could direct the reader towards a model validation exercise, which is not the focus of the paper. We think whiteboard profile summaries are an excellent example of how forecasters assimilate and summarize information into an abstract representation, which is more along the theme and inspiration of our study. Unfortunately, we do not have a photo of visual snowpack summary for our study area or study periods, so instead use one from somewhere else.

- *p7 l17-19: Could you comment on the specs of the profile locations and how they are chosen/defined to be representative for the study region? Is the number of study plots important? What number is expected to provide a representative analysis for the region (or spatial density)? Since spatial distribution is an important point of your analysis it could be worth to provide a corresponding map overview of the study region and profile locations (in particular for readers that are not familiar with the region).*

We have added a map of the study area (Fig. 2) and a discussion about the importance and challenges of configuring models to obtain a representative sample. We further describe our method and acknowledge that obtaining a representative sample of profiles is a difficult question that will vary between context. We also discuss some of the visualization challenges of small or large datasets in the Discussion. In certain forecasting contexts it could be helpful and easy to implement a map in the dashboard where groups of profiles are selected by location, but this is not necessary in the context of regional forecasting where the area is pre-defined.

- *p7 l20-25: Why is the emphasis undesired? Could you elaborate a bit with respect to what some features have high or low importance?*

We've added some examples including the fact that thin surface hoar layers

**are important but often have minimal contrast with surrounding layers in the standard palette.**

- *p8 l7-9, Table 2: Could you please add some references and comments on the simplified groups of grain types, with particular emphasis on why it would or why it would not be appropriate to summarize RG and FC as bulk layers (with respect to different types of metamorphism / underlying physical process). In your example (see e.g. Fig 2) it could appear also appropriate to add MF as bulk layer?*

**This grouping is now explained and justified in more detail. In short, SNOW-PACK has a size threshold where FC larger than 1.5 mm are called DH. Thus most FC layers in the model appear in bulk layers while most problematic FC layers appear as DH. We also cite Schweizer and Jamieson (2007) for common rules about identifying weak layers (i.e. the combined importance of grain type and size).**

**Yes our example has some thick layers of melt forms, however the colour palette places bulk layers and melt layers as equals on the visual hierarchy of importance. It could be argued to use a single hue for bulk and melt layers, however melt layers can be unique in terms of avalanche release since they are often much harder and can either bridge a weak layer or for a bed surface. Using distinct groups can help identify some of these stratigraphy patterns.**

- *p9 l11-12: Please Specify (see also comment on Figure 4). Are you displaying the percentage for a specific date (8 January 2018)?*

**We corrected our aggregation algorithm to avoid percentages greater than 100% and extended our figure caption and text description to explain it is the percentage profiles in the study area with a given layer date and grain type combination.**

- *p11 l14: Could you provide a (technical) reference for the "jitter" plot?*

**We now cite Ellis and Dix (2007) who discuss techniques for clutter reduction, including jitter.**

- *p12 l10-14: I think it would be worth to (1) mention the availability of the median values in the interactive dashboard (which are way more instructive than the figure) and (2) to comment on the spatial variability of throughout your profiles, e.g. by mentioning the standard deviation and median values for the expected depth main avalanche problems.*

**We now mention the relevance of browsing the distribution of layer depths and the fact interactive tools can help by showing summary statistics for selected features.**

- *p13 l6: Is prevalence really connected to spatial distribution (including all sample pits) or is a total measure of occurrence?*

**The CMAH defines spatial distribution of a problem as "the spatial density and distribution of an avalanche problem and the ease of finding evidence to support or refute its presence". We argue the total measure of occurrence is related to the spatial density (assuming the data is a representative sample of snowpack variability).**

- *p13 l9-10: I think it would be worth to shortly discuss why the storm slab (that previously appeared as one of the main problems) is not highlighted in the sensitivity to triggers?*

**We make it clearer that the SSI is best suited for problems related to deep weak layers (since it neglects layers within ski penetration). This also leads to our expanded discussion about using different attributes to characterize the likelihood of different avalanche problem types.**

- *p14 sec. 3.5: I find this section highly instructive to understand figures and conclusions in this paper and would like to see interactive CMAH dashboard men-*

*tioned earlier in the paper, since it e.g. provides more instructive/clear visualization than the printed terrain class visualization (Fig 5).*

**We agree with this suggestion because the interactive CMAH dashboard tool is our best example of applying visualization principles, so we now highlight the tool earlier in the paper and frame each individual visualization as a component of the tool. The dashboard now receives acknowledgment throughout Sect. 3.**

*The figures generally appear clear but would benefit from a (more) self-sufficient description (by e.g. referring to the interactive dashboard where applicable and indicating the specific date in all plots/captions (where applicable, e.g. of Fig 2 (right) and Fig 3,4, 5, 6, 7, 8 )):*

**We provide a more complete description of the data included in each figure caption.**

- *Figure 1: Could you enhance this Figure / increase readability and describe which information is given in the generalized snow profile or alternatively provide a manual profile of the study region/time as reference (additionally a map view of modelled study plots could be beneficiary here, see comment below)?*

**We have replaced the image with a clearer image from a different date and provide a more detailed description of how to interpret the information. We have also added a map of the study area (Fig. 2).**

- *Figure 2: Please indicate/describe somewhere (text and caption) what the difference between the left (timeline of stratigraphy) and right (hardness vs depth for a specific date (which?)) plot are?*

**The timeline and stratigraphy profiles are now explained in greater detail in the caption.**

- *Figure 3: It would be helpful somehow give an overview of the study area and where the profiles are simulated.*

  **The new study area figure (Fig. 2) will help readers unfamiliar with our study area. Maps can easily be added to interactive tools to select profile locations, which could be useful in some contexts, but is not critical to for this example to answer the questions of the CMAH.**

- *Figure 4, Figure 7: How can 100% be exceeded (e.g. Nov 24/23 and Oct 22)? Please double check your scale or explain (see comment above concerning prevalence). In the caption - persistent grain types or grain types associated to potential/persistent weak layers (e.g. these are mentioned as weak layer in the dashboard, please double check for the sake of consistency)?*

  **We have corrected the algorithm and checked for consistent terminology between the text and caption.**

- *Figure 7: ...shows spatial distribution... Is it really the spatial distribution in this case (like e.g. Fig 5), or rather a total measure of occurrence (see comment above). Would it be beneficial/feasible to also use the size scaling of Fig 6 allowing for a visual comparison/connection btw. the Figures?*

  **We argue the total measure of occurrence is related to the spatial density, and serves as an approximate indication if a problem is widespread/specific/isolated in the terrain. We have added the size scaling to make a connection between the figures.**

- *Figure 8: Why are IF not specified/displayed in the dashboard?*

  **The legend hid IF because they were not present in any of the profiles, so we manually added IF to the legend for this figure.**

**RESPONSE TO RC3 (Michael Warscher)**

*In their manuscript "Enhancing the operational value of snowpack models with visual-
ization design principles", the authors present the application of different visualization
design principles in the domain of avalanche forecasting using data from the widely
used model SNOWPACK.*

*GENERAL COMMENTS*

*The manuscript is technically very well written, as well as easily readable and under-
standable. I list some general comments and specific remarks in the following.*

- *I fully understand and appreciate the usefulness of the presented visualizations
  and their simplification and aggregation character, however, I still would like to
  additionally see some conventional map plots at the top-level of the dashboard.
  This would be very helpful to get an overview of the domain and the spatial dis-
  tribution of specific snow characteristics and avalanche problems. Examples of
  such visualizations are presented in e.g. Morin et al. 2020. Your aggregated
  plots would be a perfect summarizing and aggregating approach in a second
  visualization step.*

  **We agree maps are a very valuable visualization tool. Following the CMAH
  workflow, the first step is establishing operational objectives and spa-
  tiotemporal context. Our context is a regional forecast, hence the study
  area is pre-defined at this stage. We added a map of the study area to help
  readers establish this context (Fig. 2). However, once the context is es-
  tablished there is no need for maps to answer the CMAH questions. We
  agree that integrating maps into the dashboard could be useful for other
  contexts, but don't think it is essential to illustrate our main message of
  applying visualization design principles.**

- *In my opinion, the most important missing approach in the presented framework is the implementation of validation data. You state in different parts of the manuscript that practitioners lack trust in the integrity of model data. They won't gain any if they do not see the model performance at some validation points at a glance in the operational setup or at least in some hindcast simulations. I think some of the presented visualizations are perfectly suited to include observed validation data. You could simply include an interface to integrate measured snow profiles and plot them right into your visualizations as single highlighted data points or in the best case, somehow link them to their respective model grid point (this way, they could be included in all your visualizations, even the "sorted-by-depth" ones). I understand that it could be complicated to do this in a visually attractive way, but I think it would be well worth the effort.*

  **We agree this would be a very valuable addition, but one that deserves special attention that goes beyond the scope of our focus on emphasizing the potential value of user-focused designs. However, we added a dedicated section called "Sect. 4.2 steps towards operational implementation" where we emphasize the importance and potential approaches for designing visualizations that support validation and understanding uncertainty.**

- *While I very much like the presentation of your new color profiles, I am kind of torn as they are very much tailored to previous existing expert knowledge (potential weak layer = surface/depth hoar = highly visible) and is not very generic. Of course, this is very useful to detect the targeted wind slab avalanche problems, but what about other common avalanche problems (e.g. wet-snow avalanches). Are they also clearly visible in your visualizations? Regarding this remark, - if feasible - it would be very beneficial for the manuscript to include an additional example for a very different avalanche situation in the same domain.*

  **Additional figures in the Appendix now show the visualizations for different snowpack conditions. The colour palettes are heavily tuned to iden-**

tifying persistent slab problems largely due to operational feedback from forecasters who see this as the potential greatest added value from snowpack models. We've added some discussion acknowledging similar design principles could be applied to identify attributes associated with other problem types (such as visualizing weather data to identify wind slab problems or snow temperature to identify wet problems). The colour palettes we propose are simply examples of how perceptual considerations should be leveraged to direct our attention towards features of interest. We added an Appendix figure with examples of the visualizations for different snowpack conditions throughout the season.**

- *I don't see the point of having so many words printed in italic letters even if they refer to specific technical terms. I think this is not necessary here and they could all just be changed to normal fonts.*

  **We removed the italicized terms.**

- *As the manuscript provides a technical report of the application of a visualization concept, it would be very beneficial to add information about the minimum requirements for a snowpack model in terms of resolution, simulation variables and output that is needed to feed the visualization software and dashboard. It is obvious that the software was developed for the use with SNOWPACK as a well-known and established snow (layer) model, but it would be interesting to read some more technical details about input requirements and portability.*

  **We address this in two ways. One, we add more description to emphasize the importance of configuring a snowpack model to capture variability within a region, which depends on context and requires model expertise. Second, our discussion of the algorithm design level in Sect. 4.1 now includes issues that arise with large or small datasets.**

- *It would also be useful to include some more variables displayed in your visualiza-*

*tions, e.g. depth profiles of snow temperatures or snow density which might also be useful for avalanche practitioners and should be provided by the SNOWPACK model.*

**As explained above, we did not intend to propose an optimized forecasting tool. However, we have added more acknowledgement that additional attributes are likely needed to identify and characterize different avalanche problem types.**

• *The user survey presented in section 4 is very little explained and far from being representative, so you should consider removing the section and just move the last sentence of the section to your conclusions.*

**We removed the survey and clearly emphasized the need for more user testing in Sect. 4.2.**

• *I have two other comments, which might well be beyond the scope of this paper, but could be a useful addition for the future development of the presented approach: In addition to the above-mentioned validation data, it would be very useful to provide a framework for ensemble simulations including uncertainty measures. The implementation of visualizations for multi-model results and corresponding model spreads and uncertainties (ensemble model outputs from e.g. different initial conditions, different meteorological forcing data, and different snow pack models) would be a logical and highly valuable (or even necessary) next step for the application of snowpack models in real-world operational avalanche forecasting settings (similar to NWP). You should add this somewhere in your conclusions. Another helpful addition for avalanche forecasters and practitioners would be the visualization of the meteorological input in your visualization framework, e.g. wind speed and gusts, (min./max./mean.) air temperature, liquid/solid precipitation, SW/LW radiation, all separated for elevation and aspect bands and sectors (of course depending on resolution and origin of the gridded meteorolog-*

*ical forcing, domain size, etc.).*

**Thank you for sharing these ideas. We agree there is great potential for applying visualization principles to combine various types of weather, snowpack, and avalanche data. We added a specific section (Sect. 4.2) to address some of these potential next steps and hope this paper serves as a foundation for how visualization approaches can help advance both the research and operational use of snowpack models.**

*SPECIFIC COMMENTS*

- *P. 1, L. 8/9: Rephrase the sentence "Examples of visualizations that support these tasks are presented and follow established perceptual and cognitive principles from the field of information visualization.", to e.g. "Examples of visualizations that support these tasks and follow established perceptual and cognitive principles from the field of information visualization are presented."*

  **Done.**

- *P.1, L. 18 and others: Regarding the term "workstations". Maybe Benjamin et al. 2019 labelled the development of software, more powerful computers and more available model and observation data as kind of mythical "workstations", I would prefer just to call it what it is, namely more powerful computers, more data, and better visualization tools that gradually developed in NWP and of course in all other fields.*

  **Workstation was the term used in Benjamin et al (2019) meteorology paper, but to be more specific we replaced that term with either "visualization tool" or "workflow" throughout the manuscript as appropriate.**

- *P. 2, L. 7 and others: update citation Morin et al., is published now.*

  **Done.**

- *P.5, L. 10: "as hardness profiles" instead of "as a hardness profiles"*

  **Done.**

- *P. 6. Fig. 1: Do you have a version with better image quality available? The figure is very hard to read. However, I would suggest to remove Fig. 1 anyways as it does not contain important information in the context of the manuscript. If you decide to keep it, you should add some more information to the manuscript explaining what the reader is supposed to see in the figure.*

  **We replaced with a higher quality figure. We think the whiteboard profile summary is an excellent example of how forecasters assimilate and summarize information into an abstract representation, and was a major inspiration for some of our designs. We add more description of these profiles and explain more in the text why it was included in the paper.**

- *P. 9, Fig. 2, x-axes right panel-plots: Please add explanation for the hardness abbreviations and a "hardness" x-axis label. It becomes clear from the text, but should be included in the figure or at least in the figure caption. That also holds for the hardness test abbreviations (F, 4F, 1F, P, K) which are clear for an avalanche practitioner (fist, 4 fingers, 1 finger, pencil, knife), but the article might be interesting for a broader (snow) scientific audience. Please add explanations.*

  **Done.**

- *P. 9, L. 15: "Herla et al., in preparation" should be removed if not already published by now.*

  **Removed citation as this work is not published yet.**

- *P. 10, Fig. 3: Even if it is clear when reading the manuscript and figure caption, I would prefer to have an arrow-type label on the x-axis (e.g. "Thinnest snowpack <-> Thickest snowpack")*

**Done.**

- *P. 11, L. 7: "way to visualize" instead of "way visualize"*

  **Done.**

- *P. 11, L. 7: "way to visualize" instead of "way visualize"*

  **Done.**

- *P. 11, L. 10: I suggest to rephrase the sentence: "Instead, using eyes to: : :", e.g. "Instead, simultaneously comparing 1D/2D visualizations: : :"*

  **Done.**

- *P. 13, Fig. 6, caption: "slab" instead of "slabs"*

  **Done.**

- *P. 14, Fig. 7: Labels "Sep 30" and "Sep 23" overlap, please solve this issue.*

  **Removed overlapping labels.**

- *P.14, L. 2: Please use italic here ("Tableau") as this seems to be the name of a commercial software developing company. Just a comment: it would be very beneficial if you would develop the dashboard in R or a similar open source programming language, as you have already done with the visualizations. This would foster the use of your very useful software by different target groups.*

  **Tableau is a fast and easy visualization prototyping software, while R is very limited in terms of interactive visualization. While similar designs can be done with open source visualization libraries (e.g. D3), they tend to be more rigid and are better suited for later stages in the design process. We intend to make more of our tools openly available when they get to that stage.**

- *P. 15, Fig. 8: Could you provide a screenshot with better quality? The very useful dashboard is kind of hard to acknowledge here.*

  **We increased the resolution of the figure.**